# The membrane-proximal external region of human immunodeficiency virus (HIV-1) envelope glycoprotein trimers in A18-lipid nanodiscs

Yi Qi[1,11], Shijian Zhang[2,3,11], Kunyu Wang[4,5], Haitao Ding [6], Zhiqing Zhang[2,3], Saumya Anang[2,3], Hanh T. Nguyen[2,3], John C. Kappes[6,7], Joseph Sodroski [2,3,8,12] ✉ & Youdong Mao [1,4,5,9,10,12] ✉

During human immunodeficiency virus (HIV-1) entry, the metastable pretriggered envelope glycoprotein (Env) trimer ((gp120/gp41)₃) opens asymmetrically. We present cryo-EM structures of cleaved asymmetric Env trimers in amphipol-lipid nanodiscs. The gp41 membrane-proximal external region (MPER) could be traced in Env protomers that remained close to the nanodisc despite Env tilting. The MPER interacts with the gp120 C-termini and gp41 α9 helices at the base of the Env trimer. MPER conformation is coupled with the tilt angles of the α9 helices, the helicity of the gp41 heptad repeat (HR1$_N$) regions, and the opening angles between the protomers of the asymmetric trimers. Our structural models explain the stabilizing effects of MPER integrity and Env proteolytic maturation on the pretriggered Env conformation. Superimposed on the asymmetry of the Env protomers, variation in the glycans at the trimer apex creates substantial structural heterogeneity in the V2 quaternary epitopes of difficult-to-elicit broadly neutralizing antibodies.

The envelope glycoprotein (Env) trimer [(gp120/gp41)₃] mediates the entry of human immunodeficiency virus (HIV-1), the cause of acquired immunodeficiency syndrome (AIDS), into host cells[1,2]. In infected cells, proteolytic cleavage of a gp160 precursor produces the mature Env, which is preferentially incorporated into virions[3,4]. Binding to the receptors, CD4 and either CCR5 or CXCR4, triggers conformational changes in Env that ultimately result in the fusion of the viral and target cell membranes[1,2]. The pretriggered (State-1) conformation (PTC) of Env is initially driven by CD4 binding into an asymmetric, default intermediate (State-2) conformation and then into the full CD4-bound (State-3) conformation[5–10]. During this process, the Env trimer "opens" and previously buried gp41 ectodomain elements form an extended heptad repeat (HR1) helical coiled coil that translocates the hydrophobic fusion peptide closer to the cell membrane[8–12]. CCR5/CXCR4 binding by gp120 induces further conformational changes in

Env that result in the formation of a stable six-helix bundle composed of the HR1 and HR2 gp41 helices[13,14]. Six-helix bundle formation brings the viral membrane (with the gp41 transmembrane (TM) region) and the target cell membrane (with the gp41 fusion peptide) together, resulting in membrane fusion and virus entry[2,13–15].

The mature Envs of primary HIV-1 strains mainly occupy the PTC but, depending on PTC stability, spontaneously assume State 2/3-like conformations to various extents[5]. Desirable but inefficiently elicited broadly neutralizing antibodies (bNAbs) generally recognize the PTC[5,16]. Exceptions are the bNAbs directed against the gp41 membrane-proximal external region (MPER) that neutralize primary HIV-1 strains by binding Env after CD4 engagement[17,18]. In contrast to the bNAbs, poorly neutralizing antibodies (pNAbs) are readily elicited but recognize Env elements, including State-2/3 conformations, that are sterically inaccessible on the functional

[1]Center for Quantitative Biology, Academy for Advanced Interdisciplinary Studies, Peking University, Beijing, China. [2]Department of Cancer Immunology and Virology, Dana-Farber Cancer Institute, Boston, MA, USA. [3]Department of Microbiology, Harvard Medical School, Boston, MA, USA. [4]State Key Laboratory for Mesoscopic Physics, School of Physics, Peking University, Beijing, China. [5]Peking-Tsinghua Joint Center for Life Science, Peking University, Beijing, China. [6]Department of Medicine, University of Alabama at Birmingham, Alabama, USA. [7]Birmingham Veterans Affairs Medical Center, Research Service, Birmingham, AL, USA. [8]Department of Immunology and Infectious Diseases, Harvard T.H. Chan School of Public Health, Boston, MA, USA. [9]National Biomedical Imaging Center, Peking University, Beijing, China. [10]School of Chemical Biology and Biotechnology, Peking University Shenzhen Graduate School, Shenzhen, China. [11]These authors contributed equally: Yi Qi, Shijian Zhang. [12]These authors jointly supervised this work: Joseph Sodroski, Youdong Mao. ✉e-mail: joseph_sodroski@dfci.harvard.edu; ymao@pku.edu.cn

Env trimer[16,19]. The State-2 Env represents a default intermediate conformation that is spontaneously assumed when State 1 is destabilized by particular changes in Env or disruption of Env-membrane interactions[6,20,21]. Cleavage of the gp160 Env precursor stabilizes the PTC[22–25]. By contrast, single-residue changes in the MPER destabilize the PTC[26–30]. Current Env structures do not explain these two observations.

Structures of soluble or solubilized HIV-1 Envs have contributed to our understanding of virus entry, inhibition by small molecules, and antibody neutralization. Stabilized soluble gp140 (sgp140) SOSIP.664 Env trimers retain many epitopes for bNAbs, allowing their structural characterization[31]. To date, sgp140 SOSIP.664 trimers have failed to elicit bNAbs efficiently, perhaps due to differences in conformation and glycosylation from State-1 virion Envs[32–37]. A detailed structure of the complete ectodomain of the pretriggered (State-1) membrane Env is currently lacking. The sgp140 SOSIP.664 Env trimers are truncated after gp41 residue 664, removing the MPER, TM and cytoplasmic tail[31]. To date, the MPER and TM of Env trimers purified from membranes have been poorly ordered, precluding structure determination[25,38–42].

In this work, we purified and determined the cryo-EM structure of a cleaved membrane HIV-1 Env (hereafter referred to as Tri FPPR Env) modified by a combination of six changes that stabilize the PTC[18]. These changes, which represent polymorphisms found in natural HIV-1 variants, were empirically selected based on viral phenotypes (resistance to CD4-mimetic molecules and cold exposure) associated with the stability of the PTC. The Tri FPPR Env was complexed with a State-1-stabilizing entry inhibitor, BMS-806[5–11], and solubilized directly in amphipol A18-lipid nanodiscs[43,44]. Details of the rationale and methods of Tri FPPR Env engineering and solubilization were recently reported[18,43]. The reconstructed Env trimers were asymmetric, with one small and two large opening angles, similar to previously characterized HIV-1 Env trimers in nanodiscs[41]. Residual order in the membrane-proximal elements allowed modeling of two asymmetric classes of Tri FPPR Env trimers that revealed the gp41 MPER structure in the absence of bound antibodies. The Tri FPPR Env structures suggest that the gp41 MPERs, $\alpha$9 helices, and free gp120 C-termini interact at the base of the Env ectodomain, explaining why MPER integrity and proteolytic cleavage of the gp160 Env precursor are critical for maintaining the PTC. By comparing the structural components of the protomers within the asymmetric Env trimers, we identify correlated conformational transitions that lead to trimer opening. We also uncover glycan heterogeneity at the Env trimer apex that may contribute to the evasion of broadly neutralizing V2 quaternary antibodies.

## Results

### Purification and characterization of the cleaved full-length Tri FPPR Env

The full-length Tri FPPR Env, which contains changes that favor the PTC of membrane Env[18], was expressed in A549 human lung epithelial cells (Supplementary Fig. 1a, b). Membranes prepared from these cells were used for Env purification. BMS-806, which slows Env transitions from the PTC[5,11], was added to the membranes and was present throughout the purification of the Tri FPPR Env. The Tri FPPR Env was extracted directly from the membranes with Amphipol A18[43,44], purified using the carboxy-terminal six-histidine (His$_6$) affinity tag, and subjected to counterselection with the 19b and F240 poorly neutralizing antibodies to remove uncleaved Envs and gp41 stumps that have shed gp120. Approximately 96% of the purified Tri FPPR Env in the preparation was cleaved (Supplementary Fig. 1c). Tri FPPR Env trimers were evident on a Blue Native polyacrylamide gel. The purified cleaved Tri FPPR Env was recognized by a panel of bNAbs and not by a panel of pNAbs; this pattern of antibody recognition was similar to that of the Tri FPPR Env on the surface of A549 cells[43].

### Two conformers of asymmetric Tri FPPR Env structures determined by cryo-EM

We used single-particle cryo-EM to analyze the structure of the Tri FPPR Env in A18-lipid nanodiscs. Extensive 2D and 3D analysis without imposing symmetry constraints yielded two 3D maps notable for their recognizable MPER density (Supplementary Fig. 2 and see below). These maps are designated Tri FPPR.1 and Tri FPPR.2, with average resolutions of 3.4 Å and 3.5 Å, respectively (Fig. 1, Table 1, Supplementary Fig. 3). The quality of the Tri FPPR.1 and Tri FPPR.2 maps allowed atomic modeling and refinement with accuracy to the level of the C$\alpha$ backbone trace and some large side chains (Supplementary Fig. 4a). With the exception of certain glycans in the gp120 variable regions (modifying Asn 136, 141 and 142 in V1; Asn 406 and 411 in V4; and Asn 463 in V5), the density associated with the Env glycans was apparent, allowing modeling of the peptide-proximal carbohydrate residues (For example, see Supplementary Fig. 4b). Ectodomain residues 31-662 of the Tri FPPR.1 and Tri FPPR.2 trimers share an overall topology with each other and with those of existing soluble and membrane Env trimer structures[25,38–42,45–48]. In all these structures, the gp120 subunits project from the gp41 subunits, with the C-terminal segments of the heptad repeat 1 (HR1$_C$) regions of gp41 forming a three-helix bundle. The gp41 MPER (residues 663-683), transmembrane (TM) region (residues 684-705) and cytoplasmic tail (CT) are not present in sgp140 SOSIP.664 trimers[31] and are not well resolved in available structures of detergent-solubilized Env trimers, including those reconstituted into lipid-bilayer nanodiscs[25,38–42]. The Tri FPPR.1 and Tri FPPR.2 maps both exhibited significant density corresponding to the MPER and N-terminal TM, with average resolutions of 8 Å and 6-8 Å, respectively. The CT and C-terminal portions of TM were not resolved in either the Tri FPPR.1 or Tri FPPR.2 maps.

The Tri FPPR.1 or Tri FPPR.2 Env trimers lack C3 symmetry due to differential rotation of the protomers with respect to the trimer axis. In these asymmetric trimers, the adjacent gp120 subunits exhibit the following geometric relationship: one opening angle is less than 120° and two opening angles are greater than 120° (Fig. 1a). The same pattern of asymmetry has been observed in uncleaved HIV-1$_{JR-FL}$ Env trimers and cleaved HIV-1$_{AD8}$ and AE2 Env trimers purified from solubilized membranes[25,41] (Supplementary Fig. 5) despite differences in Env sequence, cleavage and preparation (Supplementary Fig. 1a).

Beginning with a C3-symmetric Env trimer, the asymmetric AD8, AE2, Tri FPPR.1 and Tri FPPR.2 Env trimers can be derived by rotational shifts of Protomers 2 and 3 about the trimer axis (Supplementary Fig. 6a, Table 2). In all these cases, Protomers 2 and 3 shift in a counterclockwise direction when the Env trimer is viewed from the perspective of the viral membrane. The shift in Protomer 3 exceeds that of Protomer 2, resulting in asymmetric Env structures in which the opening angle between Protomers 1 and 3 is <120° and the other two opening angles are >120°. This pattern of asymmetric Env opening has been suggested to assist the binding of two CD4 molecules, which are required for HIV-1 entry[41,49]. The shift in the gp41 subunit of Protomers 2 and 3 exceeds that of the gp120 subunit. This observation is consistent with changes in gp41 driving the shift, with the gp120 subunits following suit. In this manuscript, we adhere to the previously published designations of the asymmetric Env protomers and chains[41] (Fig. 1a); in this scheme, the smaller opening angle is between Protomers 1 and 3.

We note that the observed trimer asymmetry is not explained by non-uniform occupancy of BMS-806 in the previous study[41] or in the Tri FPPR Envs (Supplementary Fig. 4c).

### Structures of the Tri FPPR.1 and Tri FPPR.2 Envs

The structures of the Tri FPPR.1 and Tri FPPR.2 Env protomers were compared (Fig. 2). The gp120 subunits exhibited similar folds, resembling those observed in previously reported Env trimer structures[25,38–42,45–48]. When the gp120 subunits were aligned, the interprotomer C$\alpha$ RMSD was 0.63-0.75 Å for Tri FPPR.1 and 0.71-0.88 Å for Tri FPPR.2. Conformational differences among the three protomers were evident in the gp41 fusion peptide, fusion peptide-proximal region (FPPR), N-terminal heptad repeat (HR1$_N$) and $\alpha$9 helices. As has been previously seen for the HIV-1$_{AD8}$ and AE2 Env trimers[41], the configuration of these gp41 elements differed among the protomers of the asymmetric Tri FPPR trimers. Notably, helical HR1$_N$ regions are present in Protomers 1 and 2, whereas HR1$_N$ is a loop in

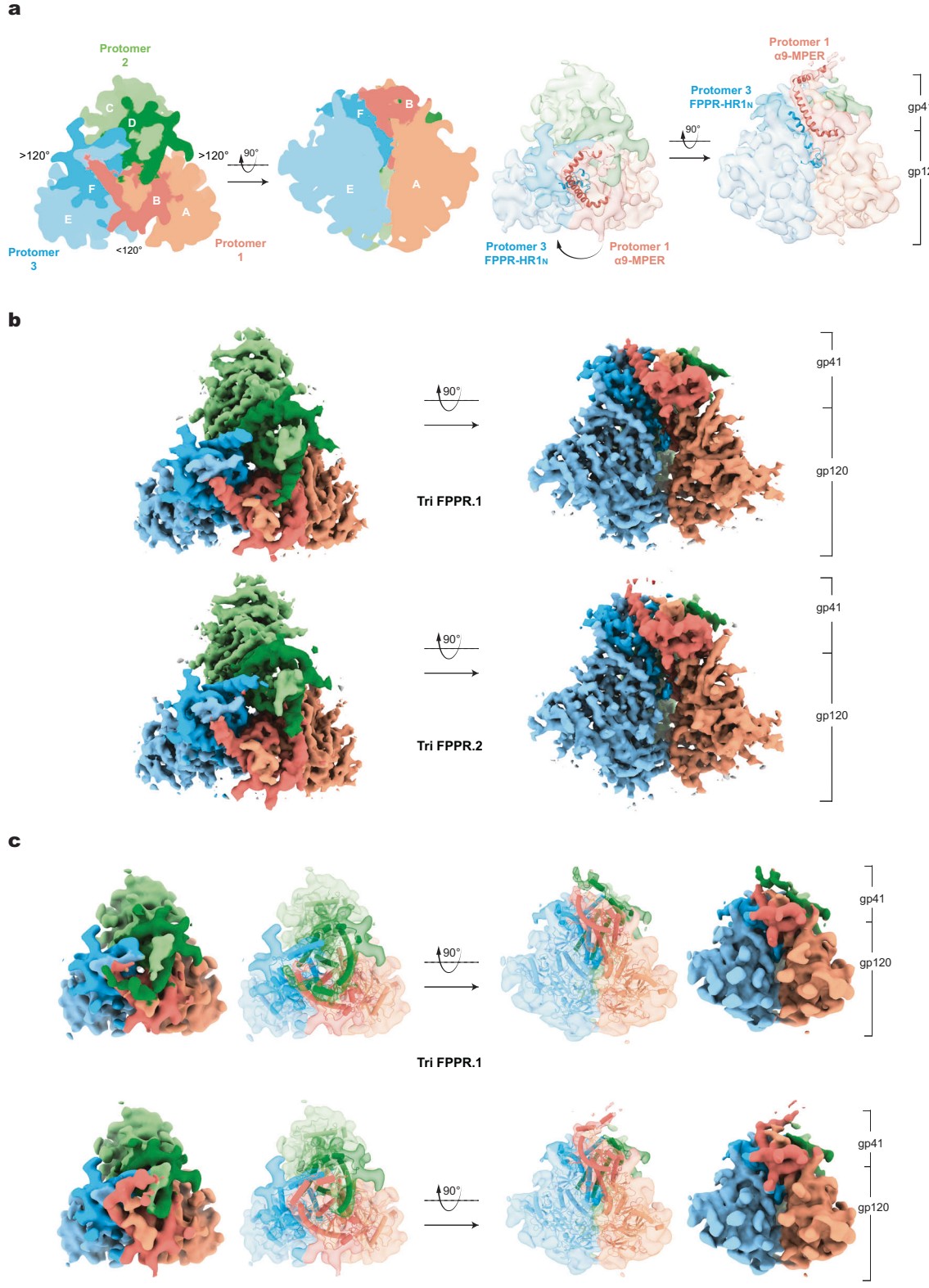

**Fig. 1 | Structures of the Tri FPPR.1 and Tri FPPR.2 Env conformers. a.** The two schematic diagrams on the left indicate the nomenclature of the gp120 chains (chains A, C and E) and gp41 chains (chains B, D and F). Except where noted, the color scheme for the Env protomers and subunits will be used throughout the manuscript. The schematic diagram on the left is a view down the Env trimer axis, from the perspective of the membrane of the virus or expressing cell. The opening angle between Protomer 1 and Protomer 3 is less than 120 degrees in these asymmetric trimers. The schematic diagram on the right is a side view, with the gp41 subunits at the top and the gp120 subunits at the bottom of the images. On the right, two 5 Å-filtered Tri FPPR.2 density maps are shown, from the same perspectives used for the schematic diagrams. The α9 helix and MPER from Protomer 1 (red ribbons) interacts with the FPPR-HR1$_N$ structures (blue ribbons) of the clockwise protomer (Protomer 3), when viewed from the perspective of the viral/cell membrane. **b** The Tri FPPR.1 and Tri FPPR.2 density maps are colored and shown from the same perspectives as in (**a**). **c** The Tri FPPR.1 and Tri FPPR.2 models are shown, fitted into the respective maps that were filtered to 5-Å resolution.

**Table 1 | Cryo-EM data collection, refinement, and validation statistics**

| | Tri FPPR.1 | Tri FPPR.2 |
|---|---|---|
| **Data collection and processing** | | |
| Magnification | 105,000 | 105,000 |
| Voltage (kV) | 300 | 300 |
| Electron exposure (e/Å) | 51.7 | 51.7 |
| Defocus range (µm) | −0.7 to −2.2 | −0.7 to −2.2 |
| Pixel size (Å) | 0.825 | 0.825 |
| Symmetry imposed | C1 | C1 |
| Initial particle images (no.) | 1,540,890 | 1,540,890 |
| Final particle images (no.) | 57,088 | 50,646 |
| Map resolution (Å) | 3.4 | 3.5 |
| FSC threshold | 0.143 | 0.143 |
| Map resolution range (Å) | 1.8–8.0 | 1.9–8.0 |
| **Refinement** | | |
| Initial model used (PDB code) | 8FAE | 8FAE |
| Model resolution (Å) | 3.7 | 3.8 |
| FSC threshold | 0.143 | 0.143 |
| Map sharpening B factor (Å²) | 0.0 | 0.0 |
| Model composition | | |
| Non-hydrogen atoms | 18,127 | 17,991 |
| Protein residues | 1884 | 1883 |
| Ligands | 243 | 233 |
| B factors (Å²) | | |
| Protein | 176.59 | 176.64 |
| Ligands | 106.48 | 98.44 |
| R.m.s.deviations | | |
| Bond lengths (Å) | 0.013 | 0.012 |
| Bond angles (degree) | 1.528 | 1.524 |
| **Validation** | | |
| MolProbity score | 2.57 | 2.54 |
| Clash score | 8.34 | 8.62 |
| Poor rotamers (%) | 6.07 | 5.23 |
| Ramachandran plot | | |
| Favored (%) | 90.92 | 90.59 |
| Allowed (%) | 7.85 | 8.66 |
| Disallowed (%) | 1.23 | 0.75 |

Single-particle cryo-EM data collection and structure refinement statistics are shown for Tri FPPR.1 and Tri FPPR.2.

Protomer 3. Interprotomer contacts between $HR1_N$ and α9 apparently influence $HR1_N$ conformation. Differences in $HR1_N$ helicity correlate with the tilt angles of the α9 helices on the adjacent protomer and with the degree of order in the MPERs extending from those α9 helices (Fig. 2 and Table 2). Thus, when Env is viewed down the trimer axis from the perspective of the membrane (as in the leftmost image in Fig. 1a), the α9-MPER conformation on each gp41 protomer is related to the $FPPR-HR1_N$ structures on the clockwise protomer (Fig. 1a and Supplementary Fig. 6b).

Remarkable differences among the Tri FPPR Env protomers were observed in the map densities associated with the gp41 MPERs. The gp41 MPER extends from the carboxyl terminus of the α9 helix (~Glu 662-Asp 664) to the transmembrane (TM) region, which begins at Ile 684. For the Tri FPPR.2 Env, the entire MPER as well as six TM residues of Protomer 1 (chain B) could be modeled (Fig. 3a). The MPER of Tri FPPR.2 chain B is

composed of two helices, $MPER_N$ and $MPER_C$, separated by a short turn and flanked by α9 and TM (Fig. 3 and Supplementary Fig. 7a). Immediately following the α9 helix of chain B, a sharp 71-degree turn at residues 664 and 665 directs the $MPER_N$ helix towards the α9 helix of the adjacent protomer [Protomer 3 (chain F)]. Sandwiched between the chain B $MPER_N$ helix and chain F α9 helix is the C-terminus of chain E gp120. The $MPER_N$ helix breaks at Thr 676-Asn 677, introducing a slight turn that directs the $MPER_C$ helix to the membrane. The initial few TM residues (Ile 684-Val 689) appear to retain a helical conformation. In contrast to the map density associated with the Tri FPPR.2 Protomer 1 (chain B), the Tri FPPR.2 map density corresponding to the MPERs in Protomer 2 (chain D) and Protomer 3 (chain F) was very weak; therefore, the chain D and F MPERs were not included in the Tri FPPR.2 Env model.

For Tri FPPR.1, the entire MPER of Protomer 2 (chain D) and the $MPER_N$ of Protomer 1 (chain B) could be modeled (Fig. 3). These MPER structures demonstrated strong similarity to those in Tri FPPR.2:

1. The chain D MPER of Tri FPPR.1 adopts a structure similar to the helix-hinge-helix configuration of the MPER in chain B of Tri FPPR.2;
2. Sharp turns between the α9 and $MPER_N$ helices direct the latter towards the α9 helix of the adjacent protomer; the angle of this turn in the Tri FPPR.1 chain D is 74°, similar to that of the Tri FPPR.2 chain B;
3. The hinge following the $MPER_N$ helices positions $MPER_C$ parallel to the α9 helix of the adjacent protomer and directs it towards the membrane;
4. The Tri FPPR.1 chain D and Tri FPPR.2 chain B $MPER_C$ segments enter the membrane in similar positions with respect to the Env ectodomain, opposite the gp41 tryptophan collar (residues 623-631) and α9 C-terminus (Glu 662-Asp 664) of the adjacent Env protomer;
5. The gp120 C-terminus is interposed between the α9 helix of the same protomer and the $MPER_N$ helix of the adjacent protomer;
6. The MPER of Protomer 3 (chain F) is poorly ordered in both Tri FPPR.1 and Tri FPPR.2 structures.

In summary, although the MPERs of both Tri FPPR.1 and Tri FPPR.2 Envs are asymmetric, each conformer retains MPER chains with highly compatible structures.

The relationship of the Tri FPPR.1 and Tri FPPR.2 Envs to the A18 amphipol-lipid nanodisc was examined (Supplementary Fig. 8). Relative to the nanodisc axis, the Tri FPPR.1 and Tri FPPR.2 ectodomains are tilted 30 and 20 degrees, respectively. When the asymmetric Tri FPPR.1 and Tri FPPR.2 Env ectodomains are aligned, the direction of their tilts with respect to the nanodiscs differs by 155 degrees. As a result, different protomers in the Tri FPPR.1 and Tri FPPR.2 Envs are positioned closer to the nanodisc surface. Notably, the proximity of the Env protomer to the nanodisc is related to the degree of order in the associated MPER. This observation implies that MPER density is preserved best when the MPER is closer to the amphipol-lipid environment of the nanodisc.

The commonalities and potential complementarity of the Tri FPPR.1 and Tri FPPR.2 maps prompted us to use a combination of these maps to model a more complete MPER structure (Fig. 4a). The resulting composite Env model suggests that interlocking gp41 MPERs, gp41 α9 helices and the gp120 C-termini from all three protomers potentially form a membrane-proximal base that supports the rest of the Env ectodomain. Single-residue MPER changes that disrupt the Env PTC[26–30] map to the $MPER_N$ helix that, in the Tri FPPR structures, abuts the gp120 C-terminus and α9 helix from the adjacent protomer (Fig. 4b).

## Env cleavage allows association of the gp120 C-terminus with the membrane-proximal base

The association of the gp120 C-terminus with the membrane-proximal base in the Tri FPPR Env structures suggested an explanation for the observation that proteolytic cleavage of the gp160 Env precursor stabilizes the PTC[22–25]. To investigate the potential contribution of Env precursor cleavage to the formation of the hypothesized membrane-proximal base, we compared the structures surrounding the Env cleavage site in the uncleaved Env(-)[25] and

**Table 2 | Parameters associated with cleaved, solubilized HIV-1 Env trimers**

| HIV-1 Env Parameter | AD8 | | AE2 | | Tri FPPR.2 | | Tri FPPR.1 | |
|---|---|---|---|---|---|---|---|---|
| Solubilization and preparation | SMA Non-crosslinked BMS-806 | | SMA Crosslinked BMS-806 | | A18 Non-crosslinked BMS-806 | | A18 Non-crosslinked BMS-806 | |
| Interprotomer opening angle (degrees)* | gp41 | gp120 | gp41 | gp120 | gp41 | gp120 | gp41 | gp120 |
| Protomers 1 and 2 | 123.6 | 123.0 | 124.3 | 122.8 | 125.6 | 124.5 | 123.2 | 123.0 |
| Protomers 2 and 3 | 129.2 | 124.9 | 128.2 | 124.4 | 125.1 | 122.6 | 128.7 | 125.1 |
| Protomers 1 and 3 | 107.3 | 112.1 | 107.8 | 112.7 | 108.0 | 113.0 | 107.8 | 112.0 |
| Shift in protomer angles from a C3-symmetric trimer (degrees) | gp41 | gp120 | gp41 | gp120 | gp41 | gp120 | gp41 | gp120 |
| Protomer 2 | 3.6 | 3.0 | 4.2 | 2.9 | 6.3 | 4.5 | 3.4 | 3.0 |
| Protomer 1 | 0 | 0 | 0 | 0 | 0 | 0 | 0 | 0 |
| Protomer 3 | 12.7 | 7.9 | 12.2 | 7.3 | 12.0 | 7.0 | 12.2 | 8.0 |
| gp41 HR1$_N$ conformation | | | | | | | | |
| Protomer 2 (chain D) | Helix | | Helix | | Helix | | Strong helix | |
| Protomer 1 (chain B) | Helix | | Helix | | Helix | | Weak helix | |
| Protomer 3 (chain F) | Unmodelled disorder | | Loop | | Loop | | Loop | |
| Tilt of α9 helix: Centroid distance (Å) from chain F | | | | | | | | |
| chain F | 0 | | 0 | | 0 | | 0 | |
| chain D | 1.5 | | 1.2 | | 0.9 | | 1.4 | |
| chain B | 2.2 | | 2.0 | | 1.3 | | 1.1 | |
| MPER density (last resolved gp41 residue) | | | | | | | | |
| Protomer 3 (chain F) | No connectivity (657) | | No connectivity (661) | | Weak density (662) | | Weak density (662) | |
| Protomer 2 (chain D) | No connectivity (657) | | No connectivity (661) | | Weak density (662) | | Complete trace (685) | |
| Protomer 1 (chain B) | Strong connectivity but no MPER order (657) | | Strong connectivity but no MPER order (664) | | Complete trace (689) | | Partial trace (669) | |

Parameters associated with the Tri FPPR.1 and Tri FPPR.2 Env trimer structures are compared with those of other cleaved, solubilized Env structures, AD8 (PDB ID: 8FAD) and AE2 (PDB ID: 8FAE)[41].
*The interprotomer opening angles were measured in glycan-free Env trimer all-atom models using PyMOL (Schrödinger LLC) with the "angle_between_domains" command, selecting the corresponding subunits from the two protomers forming the interface.

cleaved AE2[41] and Tri FPPR Env trimers (Fig. 5). As the surface-exposed cleavage site is disordered to various degrees in these Env structures, we measured the linear distance (LD) between the last resolved gp120 residue and the first resolved gp41 residue in each protomer of these Env trimers (Fig. 5a). LD represents a minimum estimate of the actual distance that must be traversed by the unresolved peptide encompassing the Env cleavage site, as steric barriers are not accounted for. If $N$ represents the number of amino acid residues available to span the linear distance LD, LD/$N$ represents the average span of an amino acid residue required to bridge the unresolved gap, in the absence of Env cleavage. LD/$N$ cannot exceed the physical length limit for an amino acid residue (~3.4 Å) without Env cleavage having occurred. Both LD and LD/$N$ were higher in the cleaved Envs than in the uncleaved Env(-) (Fig. 5a) (Mann-Whitney test, P = 0.02), indicating that Env cleavage results in the movement of one or both newly created termini away from each other. LD/$N$ strongly correlated with the last resolved gp120 C-terminal residue (Fig. 5b, left panel), but not significantly with the first resolved gp41 N-terminal residue (Fig. 5b, right panel). This relationship is consistent with Env cleavage allowing the gp120 C-terminus to achieve a more ordered conformation. We note that the LD/$N$ values varied among the cleaved Envs and even among the Tri FPPR protomers, observations that have several implications. First, the highest LD/$N$ values far exceeded the amino acid physical limit, confirming that Env cleavage must have occurred to allow the observed gp120 C-terminal conformations in these Tri FPPR Env protomers. Second, the highest LD/$N$ values were observed for the gp120 glycoproteins in chain C of Tri FPPR.1 and Chain A of Tri FPPR.2, whose C-termini potentially interact with the well-ordered MPER$_N$

helices of their respective protomers (Fig. 5c). Tri FPPR Env protomers with lower LD/$N$ ratios and less well-resolved gp120 C-termini have less ordered MPERs. Third, in the AE2 or sgp140 SOSIP.664 trimers, where the MPERs are completely disordered or absent, respectively, the gp120 C-termini were resolved only to Val 505, a residue that is positioned similarly in all cleaved Env trimer structures. These observations support a mechanism whereby Env cleavage frees the gp120 C-terminus, allowing the terminal six residues to interact with the cognate MPER, thereby stabilizing the membrane-proximal base and the pretriggered Env conformation (Fig. 5d).

**Structural correlates in cleaved, solubilized HIV-1 Env trimers**
The HIV-1$_{AD8}$, AE2, Tri FPPR.1 and Tri FPPR.2 Env trimers were solubilized from cell membranes and prepared in different ways (Supplementary Fig. 1a), yet share a common asymmetric architecture. We compared these structures to identify generalizable relationships between specific structural parameters:

1. Env tilting in the nanodisc and protomer asymmetry. In a previous study[41], using the sole available AE2 Env structure, we noted that the Env ectodomain tilted in the SMA nanodiscs towards the interprotomer interfaces with the smaller opening angle. Examination of the Tri FPPR.1 and Tri FPPR.2 Envs indicates that this is not generalizable. Whereas the Tri FPPR.1 Env tilts towards the small opening angle, the Tri FPPR.2 Env tilts away from the small opening angle (Supplementary Fig. 8).
2. MPER order/disorder and Env protomer asymmetry. In all four Env structures (AD8, AE2, Tri FPPR.1, and Tri FPPR.2), the map density

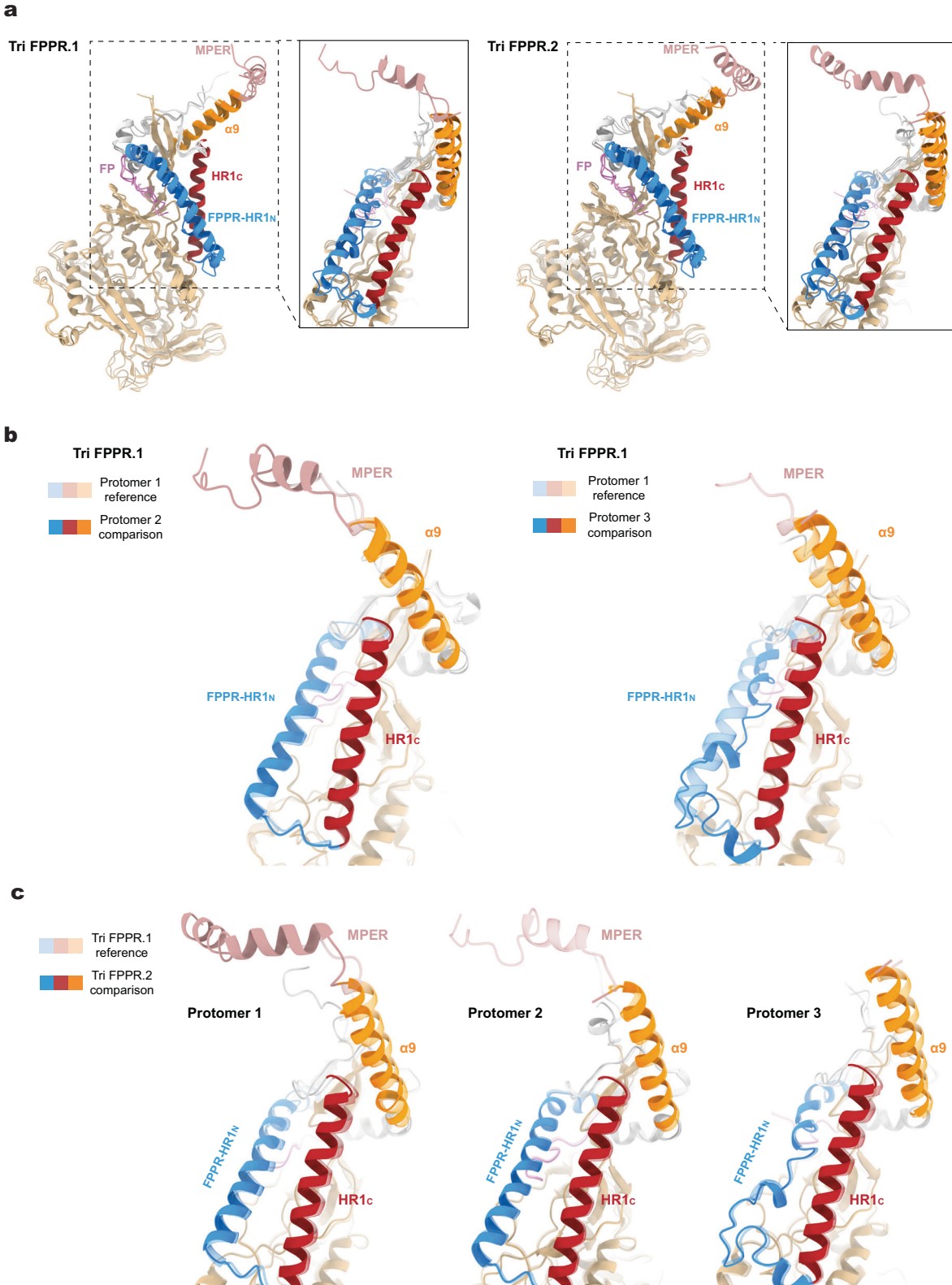

**Fig. 2 | Comparison of Tri FPPR Env protomer conformations. a–c** Molecular structures of the three protomers of the Tri FPPR.1 and Tri FPPR.2 Env models were aligned using the gp120 subunits. Conformational differences among the protomers were evident in the gp41 fusion peptide (magenta), the fusion peptide-proximal region (FPPR) and N-terminal heptad repeat (HR1$_N$) (blue), the α9 helices (orange) and the MPERs (salmon). In contrast, the gp41 HR1$_C$ helices (red) of the three protomers exhibited similar conformations. **a** The three aligned protomers of the Tri FPPR.1 model are shown in the left panel, and the three aligned protomers of the Tri FPPR.2 model are shown in the right panel. The insets show the gp41 subunits from a different perspective. **b**. In the left panel, Protomer 2 (darker shading) and Protomer 1 (lighter shading) of the Tri FPPR.1 model are compared. In the right panel, Protomer 3 (darker shading) and Protomer 1 (lighter shading) of the Tri FPPR.1 model are compared. **c** Protomers 1, 2 and 3 of the Tri FPPR.2 model (darker shading) are compared with the corresponding protomers of the Tri FPPR.1 model (lighter shading).

**Fig. 3 | Structures of the MPER in the Tri FPPR.1 and Tri FPPR.2 Envs. a** Ribbon models of the membrane-proximal structures of Tri FPPR.1 Protomer 2 and Tri FPPR.2 Protomer 1 are shown with the corresponding maps filtered to 5 Å. Note the proximity of the gp120 C-terminus to the gp41 α9-MPER$_N$ helices of the same protomer in each structure. **b** The α9 (orange), MPER (salmon) and TM (blue) of Tri FPPR.1 (left panel) and Tri FPPR.2 (right panel) are shown from the same perspective, with the turn angles between α9 and MPER$_N$ depicted. In the left panel, the α9 and MPER of Tri FPPR.1 chains D and B are compared. In the right panel, the α9 and MPER of Tri FPPR.2 chain B are shown. **c** Ribbon models of the Tri FPPR.1 and Tri FPPR.2 Env trimers are shown on the left with the subunits colored as in Fig. 1 and the MPERs at the top. In the enlarged figures on the right, the membrane-proximal base of each trimer is shown from the perspective of the membrane, looking down the trimer axis. The smallest opening angle between the gp120 subunits is at the bottom of the images. In each case, the well-ordered MPERs project from the α9 helix of their own chain towards the α9 helix of the adjacent protomer. The MPER$_C$ extends to the apex of the adjacent protomers, where the TM enters the membrane. In the Tri FPPR.1 structure, one face of the membrane-proximal base of Env is highlighted, illustrating how the gp120 C-terminus is sandwiched between the gp41 α9 helix of the same protomer and the MPER$_N$ of the adjacent protomer. This interlocking arrangement creates opportunities for stabilizing bonds among the Env protomers and subunits.

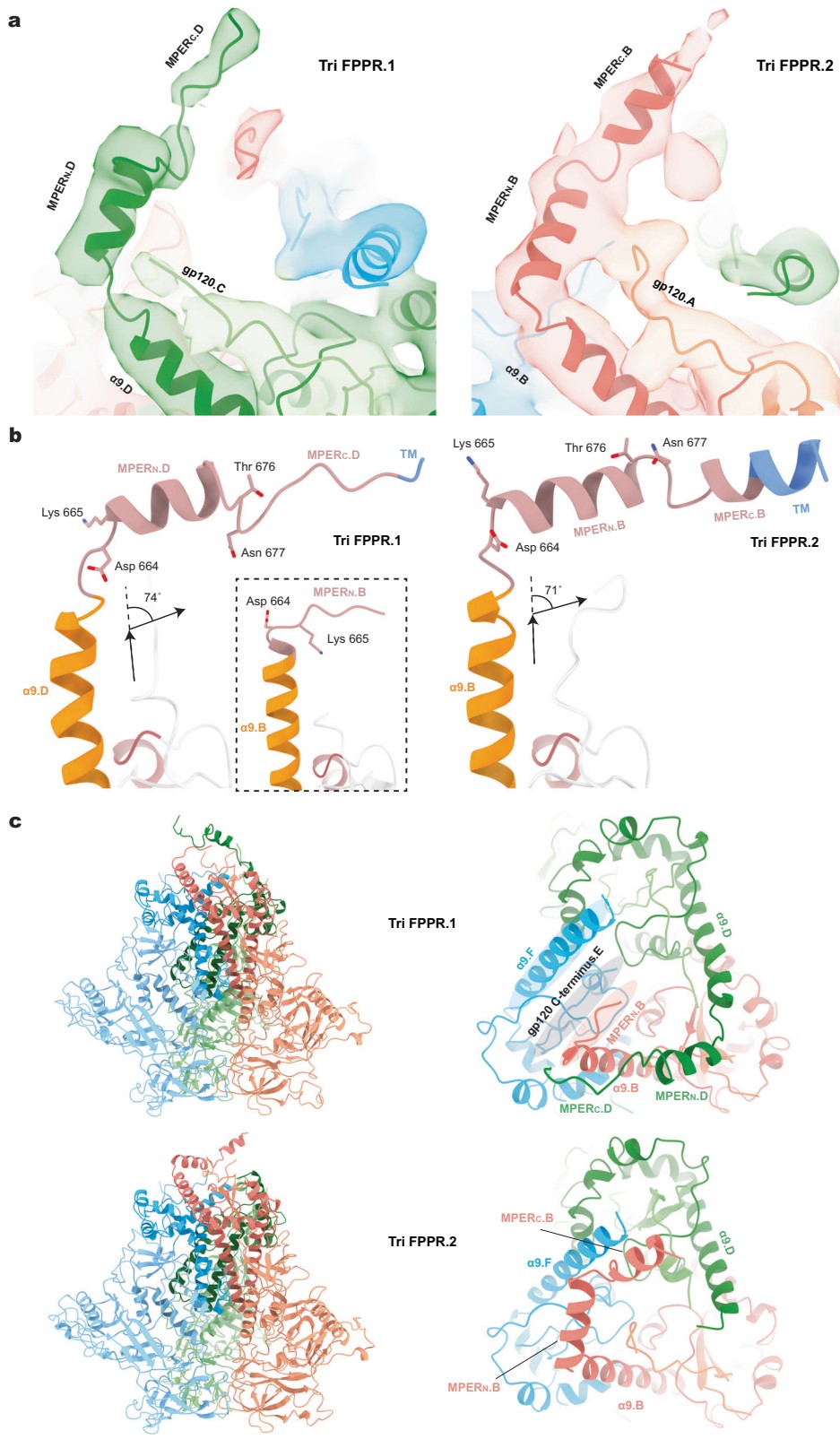

for the gp41 MPER of chain F (Protomer 3) is weak, suggesting disorder. To achieve the observed asymmetric Env trimers from a C3-symmetric trimer, the shift required in Protomer 3, particularly in gp41/chain F, is greater than that in the other protomers (Supplementary Fig. 6a). As the MPERs of the other protomers are disordered in the AD8 and AE2 Envs, we were unable to evaluate a potential correlation between MPER order and protomer asymmetry. This correlation is apparent in the Tri FPPR Env structures. The MPER density is more ordered in chain B (Protomer 1) of Tri FPPR.1 and Tri FPPR.2, and in chain D (Protomer 2) of Tri FPPR.1 (Table 2). Thus, the MPER order is associated with protomers that require smaller shifts to achieve the observed asymmetry.

**Fig. 4 | A composite model of the membrane-proximal Env base. a** To develop a more complete model of the MPER within the Env trimer structure, the 5 Å-filtered maps of Tri FPPR.1 and Tri FPPR.2 were aligned. The Tri FPPR.1 model was adjusted to fit the aligned maps, with helical fragments of the Chain B MPER-TM region from the Tri FPPR.2 model integrated accordingly, followed by refinement and modeling of the inter-helical loops to ensure structural coherence. In a similar way, the MPER$_N$ helix of chain F was modeled into appropriately located, unexplained density apparent in the lowpass-filtered Tri FPPR.2 maps. The resulting composite model is shown from the perspective of the viral membrane. The Env subunits are colored according to the scheme in Fig. 1. The composite model highlights the interprotomer interactions involving the gp41 α9 and MPER helices and the gp120 C-termini (asterisks) that potentially stabilize the membrane-proximal base of Env. The locations of the last resolved gp120 C-terminal residues (Arg 511.A, Arg 511.C, Gln 507.E) are indicated by asterisks. **b** One face of the membrane-proximal base of the Tri FPPR.2 model is shown, fitted into the unfiltered density map. To enhance clarity, disconnected bits of density with a size smaller than 8.5 Å, as measured by the largest bounding box dimension (X, Y, or Z), are not shown. Changes in MPER residues that result in the disruption of the PTC[30] are highlighted in yellow on the ribbon diagram.

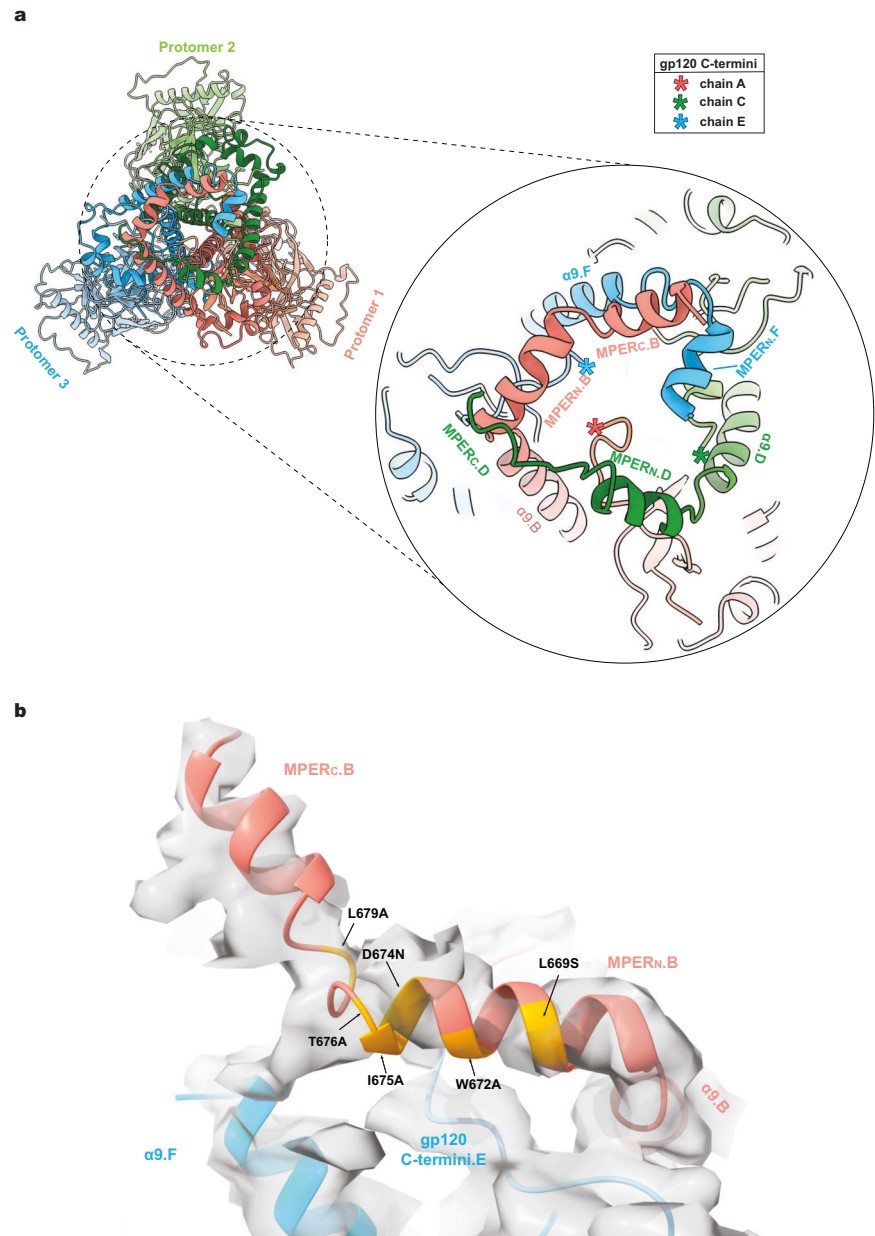

3. Connectivity to an ordered MPER and the tilt angles of the α9 helices. In each protomer, the gp41 α9 helix is immediately N-terminal to the corresponding MPER. The tilt angles of the α9 helices relative to the gp120 subunit vary among the protomers of these asymmetric Env trimers. We used the α9 helix of chain F as a reference because it is not constrained by a connection with a well-ordered MPER in any of the studied Envs. We found that the tilt of the gp41 α9 helices is inversely related to the strength of the connection with the MPER. Where MPER density is strong, indicative of an ordered MPER, the tilt angles of the α9 helices are further from those of the chain F α9 helix (Supplementary Fig. 9, Table 2). Connectivity with a stable MPER apparently restrains the α9 helices from tilting.

4. The tilt of the α9 helix and the formation of the HR1$_N$ helix on the adjacent gp41 HR1$_N$ region. Larger tilt angles of the α9 helices (with weaker MPER connectivity) correlate with the presence of helical HR1$_N$ conformations on the clockwise adjacent protomer (viewed

from the perspective of the viral membrane) (Supplementary Figs. 6b and 9, Table 2).

5. Helicity of the gp41 HR1$_N$ region and interprotomer opening angle. A strong direct correlation was observed between the helicity of the HR1$_N$ region of a given gp41 protomer and the opening angle between that protomer and the counterclockwise adjacent protomer (viewed from the perspective of the viral membrane) (Supplementary Figs. 6b and 9b, Table 2). Helix-helix packing between HR1$_N$ and α9 on the adjacent protomer should keep the interprotomer angle open.

These observations suggest a model in which Env tilting in the amphipathic copolymer-lipid nanodisc coincides with the disruption of the pretriggered state of one or two MPERs. A disordered MPER cannot maintain the conformation of the connected α9 helix, which is free to stabilize the HR1$_N$ helix on the adjacent protomer. This α9-HR1$_N$ interaction secures the opening of the interprotomer angle in the asymmetric Env trimer.

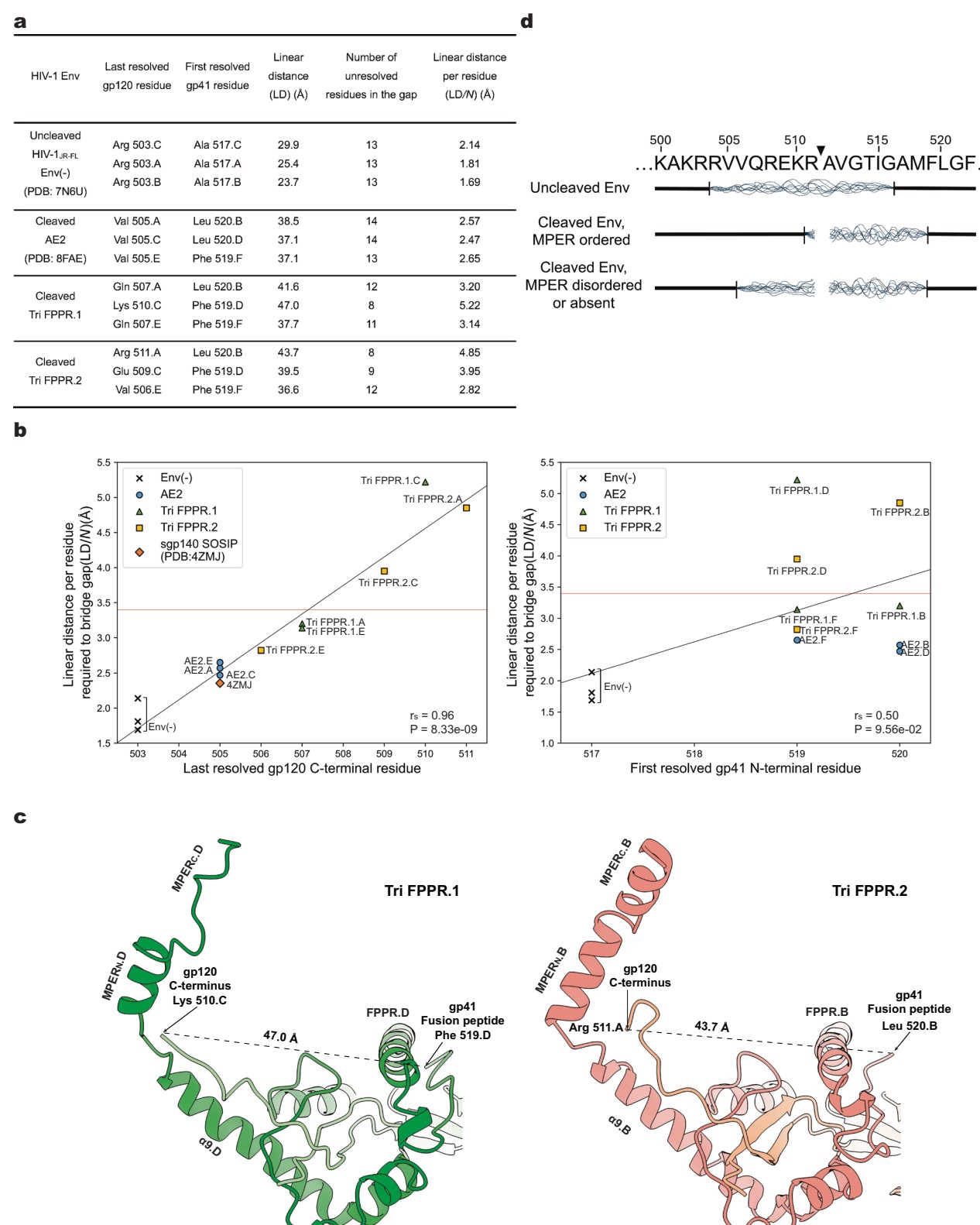

**a**

| HIV-1 Env | Last resolved gp120 residue | First resolved gp41 residue | Linear distance (LD) (Å) | Number of unresolved residues in the gap | Linear distance per residue (LD/N) (Å) |
|---|---|---|---|---|---|
| Uncleaved HIV-1<sub>JR-FL</sub> Env(-) (PDB: 7N6U) | Arg 503.C | Ala 517.C | 29.9 | 13 | 2.14 |
| | Arg 503.A | Ala 517.A | 25.4 | 13 | 1.81 |
| | Arg 503.B | Ala 517.B | 23.7 | 13 | 1.69 |
| Cleaved AE2 (PDB: 8FAE) | Val 505.A | Leu 520.B | 38.5 | 14 | 2.57 |
| | Val 505.C | Leu 520.D | 37.1 | 14 | 2.47 |
| | Val 505.E | Phe 519.F | 37.1 | 13 | 2.65 |
| Cleaved Tri FPPR.1 | Gln 507.A | Leu 520.B | 41.6 | 12 | 3.20 |
| | Lys 510.C | Phe 519.D | 47.0 | 8 | 5.22 |
| | Gln 507.E | Phe 519.F | 37.7 | 11 | 3.14 |
| Cleaved Tri FPPR.2 | Arg 511.A | Leu 520.B | 43.7 | 8 | 4.85 |
| | Glu 509.C | Phe 519.D | 39.5 | 9 | 3.95 |
| | Val 506.E | Phe 519.F | 36.6 | 12 | 2.82 |

**d**

500   505   510   515   520

...KAKRRVVQREKR▼AVGTIGAMFLGF...

Uncleaved Env

Cleaved Env, MPER ordered

Cleaved Env, MPER disordered or absent

**b**

Left plot: Linear distance per residue required to bridge gap (LD/N)(Å) vs Last resolved gp120 C-terminal residue

Legend: Env(-), AE2, Tri FPPR.1, Tri FPPR.2, sgp140 SOSIP (PDB:4ZMJ)

Data points: Tri FPPR.1.C, Tri FPPR.2.A, Tri FPPR.2.C, Tri FPPR.1.A, Tri FPPR.1.E, Tri FPPR.2.E, AE2.E, AE2.A, AE2.C, 4ZMJ, Env(-)

$r_s = 0.96$, $P = 8.33e-09$

Right plot: Linear distance per residue required to bridge gap (LD/N)(Å) vs First resolved gp41 N-terminal residue

Legend: Env(-), AE2, Tri FPPR.1, Tri FPPR.2

Data points: Tri FPPR.1.D, Tri FPPR.2.B, Tri FPPR.2.D, Tri FPPR.1.F, Tri FPPR.2.F, AE2.F, Tri FPPR.1.B, AE2.B, AE2.D, Env(-)

$r_s = 0.50$, $P = 9.56e-02$

**c**

Tri FPPR.1

MPERc.D, MPERn.D, gp120 C-terminus Lys 510.C, 47.0 Å, FPPR.D, gp41 Fusion peptide Phe 519.D, α9.D

Tri FPPR.2

MPERc.B, MPERn.B, gp120 C-terminus, Arg 511.A, 43.7 Å, FPPR.B, gp41 Fusion peptide Leu 520.B, α9.B

## Comparison of the Tri FPPR MPER with other MPER structural models

We compared our Tri FPPR structures with available structural models of the MPER without MPER-targeted antibodies (Supplementary Table 1 and Supplementary Fig. 7a–c). The structure of an MPER peptide monomer in dodecyl phosphatidylcholine (DPC) micelles has been solved by NMR spectroscopy (PDB ID: 2PV6)[50]. From the N- to the C-terminus, the MPER peptide consists of a two-turn alpha helix, a hinge, a one-turn helix and a $3_{10}$ helix (Supplementary Fig. 7a, b). Conserved hydrophobic faces of these helical structures are buried in the DPC micelle, whereas more variable hydrophilic residues near the hinge and one-turn helix are solvent exposed. We aligned the 2PV6 MPER peptide with the chain B MPER from the Tri

**Fig. 5 | Env cleavage frees the gp120 C-terminus to assume its position in the membrane-proximal base. a** The linear distances (LDs) between the last resolved gp120 C-terminal residue and the first resolved gp41 N-terminal residue in the protomers of the uncleaved HIV-1$_{JR-FL}$ Env(-) trimer (PDB ID: 7N6U)[25], the cleaved AE2 Env timer (PDB ID: 8FAE)[41], the cleaved Tri FPPR.1 Env trimer and the cleaved Tri FPPR.2 Env trimer are shown. LD was measured from C$_\alpha$ atom to C$_\alpha$ atom of the indicated gp120 and gp41 amino acid residues. $N$ is the number of residues available to fill the gap and equals the number of unresolved residues + 1. The linear distance per residue (LD/$N$) represents the average span of an amino acid residue required to bridge the unresolved gap surrounding the Env cleavage site, in the absence of cleavage. **b** The relationships between the last resolved gp120 C-terminal residue and LD/$N$ (left panel) and between the first resolved gp41 N-terminal residue and LD/$N$ (right panel) are shown. The distance spanned by an amino acid residue in a fully extended peptide strand (3.4 Å) is marked by the red horizontal line. For LD/$N$ values that exceed this physical limit, Env cleavage would be required to obtain the conformations of gp120 and gp41 observed in the structures. The degree of order in

the gp120 C-terminus, reflected in the extent of the residues resolved in the structures, strongly correlates with the LD/$N$ values. The Spearman rank order correlation coefficients ($r_S$) and two-tailed $P$ values are shown. **c** Views of the membrane-proximal region from the Tri FPPR.1 and Tri FPPR.2 Env structures are shown, illustrating the relationship of the ordered gp120 C-termini to the MPER$_N$ helices of the gp41 subunits from the same protomer. In the Tri FPPR.1 model on the left, gp120.C (light green) and gp41.D (dark green) chains are shown. In the Tri FPPR.2 model on the right, gp120.A (salmon) and gp41.B (red) chains are shown. In both cases, based on LD/$N$ values greater than 3.4 Å, the observed conformations of the gp120 C-termini in the membrane-proximal base of the Tri FPPR Env trimers could only occur after Env cleavage. **d** Env sequences and structural relationships near the gp120-gp41 cleavage site (black triangle) are summarized. Env sequences that are disordered in the analyzed structures are indicated by wavy lines. Note that the degree of order in the gp120 C terminus (residues 506-511) is dependent both on Env cleavage and on the degree of order in the MPER of the same protomer.

FPPR.2 Env (Supplementary Fig. 7b). Both structures comprise an amphipathic helical MPER$_N$ followed by a hinge that directs the C-terminal MPER segment towards the membrane. However, the boundaries of the helices and the orientations of the MPER components differ between the structures. The 2PV6 N-terminal helix breaks earlier, at Trp 672, compared with the Tri FPPR MPER$_N$ helix, which ends at Thr 676. The two structures diverge significantly around the hinge region, leading to different orientations of the C-terminal MPER elements.

By virtue of association with complete Env ectodomains or TM regions, a few unliganded MPER peptides form trimers that have been structurally characterized. Two NMR structures of MPER-TM peptides in nanodiscs or phospholipid bilayers[51,52] differ dramatically from one another; in 6E8W, the MPER-TM peptides assume a stalk-bubble conformation, whereas in 6DLN, the MPER-TM peptides assume a tripod conformation. Although both MPER-TM peptide structures contain helical elements, the helical boundaries and interhelical turns do not coincide with those of the Tri FPPR MPER$_N$ and MPER$_C$ helices, resulting in large differences in the trimeric MPER structures (Supplementary Fig. 7a, c). Likewise, in a cryo-EM structure (PDB ID: 7SC5) of a membrane SOSIP-modified Env reconstituted in nanodiscs[42], the hypothetically modeled MPER$_N$ turns from the α9 helix in a direction opposite to that seen in the Tri FPPR Envs, resulting in a dissimilar overall MPER structure. The thin stalks at the base of HIV-1 Env spikes visualized at lower resolution by cryo-electron tomography (cryo-ET)[53,54] contrast with the more substantial membrane-proximal base modeled for the Tri FPPR Env; we note that the aldrithiol (AT-2) used to inactivate the HIV-1 virions in these cryo-ET studies has been shown to contribute to destabilization of the State-1 Env conformation[53]. In summary, none of the available structural models of unliganded trimeric MPERs fully align with the Tri FPPR MPERs, underscoring the highly context-dependent nature of MPER conformation.

Variation in the overall conformation of the Env base between the Tri FPPR Env and existing antibody-free MPER structures is reflected in interprotomer distances. For example, interprotomer C$_\alpha$-C$_\alpha$ distances for Lys 665, near the N-terminal boundary of the MPER, are 24 Å for the two residues resolved in the Tri FPPR.1 model, compared with 53 Å in the 6DLN tripod, 16 Å in the 6E8W stalk-bubble and 30 Å in the 7SC5 membrane Env. Additional comparisons between the Tri FPPR composite model and available MPER peptide structures are shown in Supplementary Table 2. The Protomer 1-Protomer 2 distances in the composite model, which may be more typical of the native distances, differed significantly from those of the tripod-like 6DLN and stalk-bubble 6E8W MPER peptide structures. Although multiple variables potentially contribute to the observed differences, the connectivity of the MPER to the rest of the Env trimer ectodomain, particularly the direction of the turn arising from the α9 helix, likely influences the MPER conformation in the Tri FPPR Env.

Comparison of the Tri FPPR MPER structures with those of MPER peptides complexed with MPER bNAb fragments[55–57] revealed major differences in secondary and tertiary structure (See Supplementary Fig. 7a and

d for examples). This is consistent with the MPER bNAbs recognizing primary HIV-1 Env conformations that are downstream of those observed in the Tri FPPR Env[17,18,56].

## Glycan heterogeneity in the gp120 V1/V2 region

The densities associated with most of the glycans in the Tri FPPR.1 and Tri FPPR.2 maps were symmetrically distributed among the three Env protomers. However, 3D classification that was focused on one gp120 subunit from all protomers of the Tri FPPR Env maps yielded two major subclasses (designated gp120.1 and gp120.2) that exhibit heterogeneity in the gp120 V1/V2 region at the apex of the Env trimer. These subclasses are distributed equally between Tri FPPR.1 and Tri FPPR.2, suggesting that the variation of glycan structures is orthogonal to and uncoupled from the variation of MPER conformations. The gp120.1 map exhibits strong density associated with the V1 loop and with the Asn 130 and Asn 160 glycans, which stack against each other (Supplementary Fig. 10). The gp120.2 map exhibits stronger density associated with the glycans modifying Asn 156 and Asn 188. The representation of gp120.2 particles in the Tri FPPR Env population exceeded that of gp120.1 particles. Mixed Env trimers with at least two gp120.2 protomers were thus better represented, allowing the generation of higher-resolution maps (Supplementary Fig. 11). Permutations of the gp120.1/gp120.2 protomers, combined with the asymmetry of the Tri FPPR Envs, create substantial structural heterogeneity at the trimer apex. Such heterogeneity could diminish the generation or efficacy of V2 quaternary bNAbs, which bind the Env trimer apex in an asymmetric manner dependent on the glycans at Asn 156 and Asn 160[37,39,58,59].

## Discussion

Here we report the cryo-EM structures of two cleaved HIV-1 Env trimers solubilized directly from membranes in A18 amphipol-lipid nanodiscs. Despite the inclusion of BMS-806 and the introduction of PTC-stabilizing Env changes, the Tri FPPR Env trimers exhibited asymmetry and incompletely ordered MPERs. The Tri FPPR.1 and Tri FPPR.2 classes differed with respect to the tilt of Env in the nanodiscs and the specific protomers in which MPER order was preserved. In the protomers positioned closer to the nanodisc as a result of Env tilting, the MPERs were more ordered. Apparently, the Tri FPPR.1 and Tri FPPR.2 classes assume Env conformations that were independently generated from the membrane Env PTC during their preparation. The Tri FPPR Env structures yielded new information about the structure of the MPER and its interaction with other components of the Env trimer. We suggest that the MPERs assemble with the gp120 C-termini and the gp41 α9 helices from adjacent protomers to form a membrane-proximal base for the Env PTC (Fig. 6). This structural model explains the dependence of the metastable PTC on MPER integrity and on Env proteolytic maturation, which frees the gp120 C-termini to interact with their membrane-proximal partners.

Asymmetry in the Tri FPPR Env MPERs was linked to asymmetry in the rest of the Env ectodomain. In the Tri FPPR Envs, like other

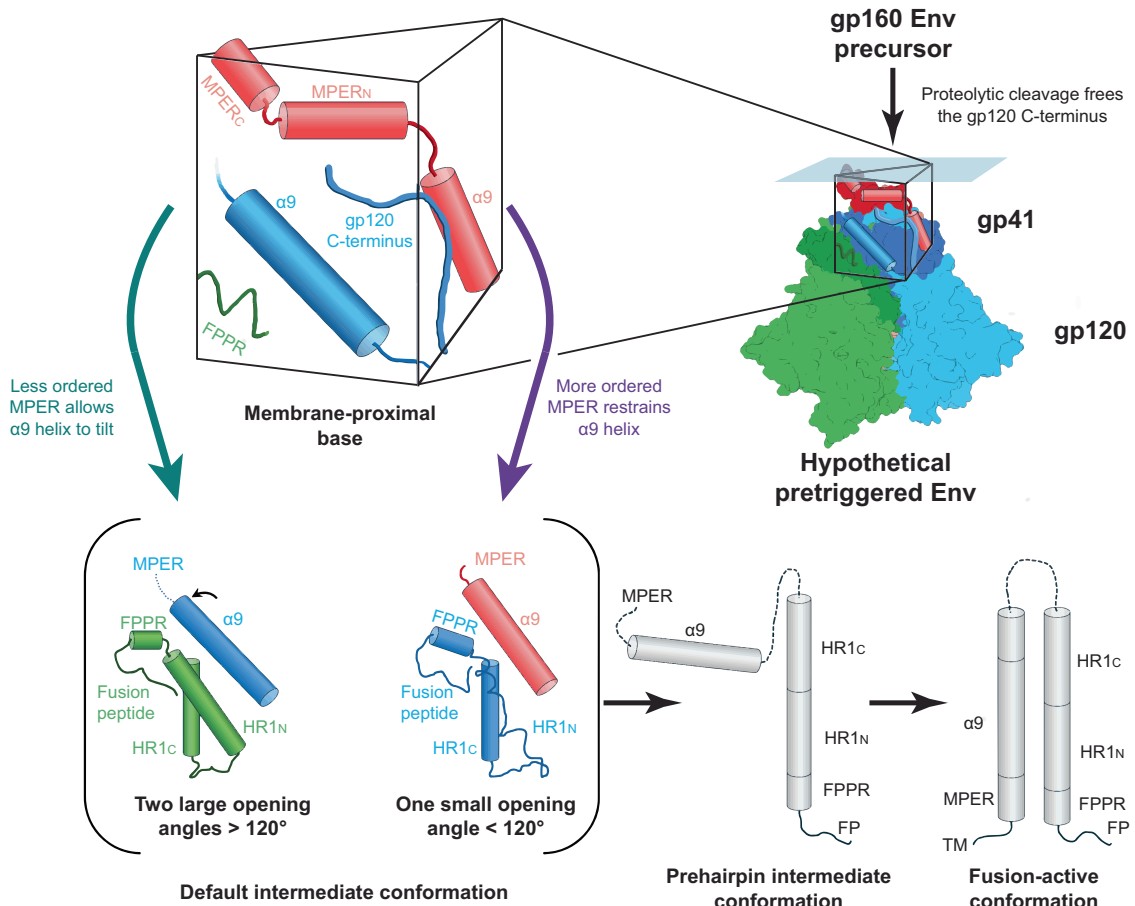

**Fig. 6 | A model for the regulation of HIV-1 Env conformation by the membrane-proximal base.** The membrane-proximal base comprises the gp120 C-termini and the gp41 MPER and α9 helices from all three Env protomers; for clarity, each of the components of the membrane-proximal base illustrated in the figure is from a single protomer. We hypothesize that an assemblage of these structural elements maintains the pretriggered (State-1) conformation of the Env ectodomain. In Env-expressing cells, cleavage of the gp160 Env precursor by the host furin protease frees the gp120 C-terminus to participate in the formation of the membrane-proximal base; this accounts for the dependence of the pretriggered (State-1) Env conformation on proteolytic maturation[22–25]. Either spontaneously or as a result of receptor binding during virus entry, the pretriggered Env on virions can assume asymmetric conformations[7–10]. If the pretriggered Env trimer is symmetric and the default intermediate conformation corresponds to that of the Tri FPPR Envs, the transition between these states would involve opening two of the interprotomer angles to greater than 120° and closing one interprotomer angle to less than 120°. In the Env protomers flanking the larger (>120°) opening angles, two conformational changes

are proposed: 1) In one of the protomers (green), the gp41 HR1$_N$ region transforms from a non-helical, as-yet-uncharacterized State-1 conformation to a helix; and 2) In the other protomer (blue), decreases in the structural order of the MPER allow the associated α9 helix to tilt and thereby stabilize the HR1$_N$ helix on the adjacent protomer. In one of the gp41 subunits (red) flanking the smaller opening angle (<120°), the α9 helix is associated with a more ordered MPER. Without the interaction afforded by the tilting of this α9 helix, the HR1$_N$ region of the other gp41 subunit (blue) flanking the smaller opening angle assumes a loop configuration. In this model, during virus entry, the transition of the gp41 HR1$_N$ region from a non-helical structure to a helix primes the formation of the long HR1$_N$-HR1$_C$ helical coiled coil in the prehairpin intermediate. Next, the interaction of the HR2 gp41 helix with the HR1 coiled coil forms the six-helix bundle, which mediates the fusion of the viral membrane (with the transmembrane (TM) anchor) and the target cell membrane (with the fusion peptide (FP)). For the prehairpin intermediate and fusion-active (six-helix bundle) conformations, only one Env protomer is depicted.

solubilized Envs, rotation of the protomers about the trimer axis results in trimers with two larger opening angles and one smaller opening angle. A helical conformation of the gp41 HR1$_N$ region in the flanking protomer is strongly correlated with a larger opening angle. We observed a relationship between the tilt of the gp41 α9 helices and α9 connectivity to an ordered MPER. These observations suggest that the MPER restrains the associated α9 helix on the same protomer from interacting with and stabilizing the HR1$_N$ helix on the adjacent protomer (Fig. 6). Thus, in these solubilized Envs, the structural stability of the MPER can influence the opening angle between the Env protomers. Similar MPER-α9-HR1$_N$ relationships in membrane Envs could explain the sensitivity of the PTC to single-residue MPER changes[26–30].

The Tri FPPR structures are attractive candidates for the default intermediate (State-2) conformation of membrane Env (Fig. 6). The default intermediate state is asymmetric and increased in occupancy when the PTC is destabilized[5–10]. The two larger opening angles could facilitate the binding

of two CD4 molecules, which is required for HIV-1 entry[49]. Transitions of the gp41 HR1$_N$ region from its as-yet-unknown conformation in the PTC to a helix would be conducive to the formation of the long HR1 coiled coil in the prehairpin intermediate (State 3). The formation of the long HR1 helix translocates the fusion peptide to the target cell membrane[8–12].

Whether the PTC of membrane HIV-1 Env is a C3-symmetric trimer is unknown. A recent study of the protomer stoichiometry of PTC-stabilizing and PTC-destabilizing Env changes suggests that conformational symmetry among the Env protomers is conducive to the maintenance of the PTC[60]. However, smFRET and antigenic analyses indicate that the conformation of HIV-1 virion Envs fluctuates between the PTC and more open conformations[5–7,61]. Spontaneously assuming asymmetric Env conformations such as those observed in the Tri FPPR Envs could assist HIV-1 immune evasion by limiting the maturation of neutralizing antibodies that recognize the PTC. For example, asymmetry amplifies structural heterogeneity at the trimer apex, where epitopes for broadly neutralizing

antibodies are located[37,39,60,61], caused by the observed variation in gp120 V2 glycosylation. Limiting Env transitions to asymmetric conformations might improve host antibody responses against the pretriggered (State-1) conformation. Conversely, driving Env prematurely into default intermediate states that decay rapidly or are vulnerable to immune clearance may have a therapeutic benefit.

## Methods

### HIV-1 Env
Compared with the HIV-1$_{AD8}$ Env from which it was derived, the Tri FPPR Env contains six changes (Q114E, A532V, I535M, L543Q, Q567K, and A582T) that stabilize the pretriggered (State-1) conformation of the membrane Env[18]. The Asp 718 (Kpn I)-Bam HI *env* fragment encoding the Tri FPPR Env was inserted into the corresponding sites of the pSVIIIenv plasmid expressing the HIV-1$_{HXBc2}$ Env and Rev proteins. Thus, the Tri FPPR Env contains a signal peptide and the carboxy-terminal portion of the cytoplasmic tail from the HIV-1$_{HXBc2}$ Env. The Tri FPPR Env has a Gly-Gly-Gly-(His)$_6$ tag at its carboxyl terminus.

### Antibodies
The poorly neutralizing antibodies 19b (against the gp120 V3 region) and F240 (against a gp41 Cluster I epitope) were obtained from James Robinson (Tulane University) and Marshall Posner (Mount Sinai Medical Center), respectively.

### Reagents
BMS-378806 (herein called BMS-806) was purchased from Selleckchem. Superflow Ni-NTA was purchased from Bio-Rad.

### Amphipol and detergent
Amphipol A18 was purchased from Cube Biotechnology. Cymal-6 was purchased from Anatrace.

### Env-expressing cell line
To establish human A549 lung epithelial cells (ATCC) inducibly expressing Rev and Tri FPPR Env, A549 cells constitutively expressing the reverse tet transactivator (rtTA) were transduced with HIV-1-based lentivirus vectors transcribing a bicistronic mRNA comprising *rev* and *env* and two selectable marker genes (puromycin and enhanced green fluorescent protein [EGFP]) fused in-frame with a T2A peptide-coding sequence[25,41]. In the transduced cells, Env expression is controlled by the Tet-responsive element (TRE) promoter and tet-on transcriptional regulatory elements. Env expression was induced by treating the cells with 2 μg/mL of doxycycline. The Env-expressing cells were enriched by fluorescence-activated cell sorting for the co-expressed EGFP marker. The polyclonal A549 cell lines inducibly expressing the Tri FPPR Env (Kappes laboratory number D1676, herein designated A549-Tri FPPR) were grown in DMEM supplemented with 10% fetal bovine serum (FBS) and penicillin-streptomycin. All cell culture reagents were purchased from Life Technologies.

### Env production and purification
A549-Tri FPPR cells were induced to express Env by culturing for 48 h in medium containing 2 μg/ml doxycycline. The cells were then incubated with 5 mM EDTA in 1× PBS at 37 °C until the cells detached from the tissue culture plates. The cells were pelleted, resuspended in 1× PBS, and centrifuged at $1500 \times g$ for 10 min. For all subsequent procedures, 10 μM BMS-806 was added to all buffers. The cell pellet was resuspended in homogenization buffer (10 mM Tris-HCl [pH 7.4], 250 mM sucrose, 1 mM EDTA and a cocktail of protease inhibitors [Roche Complete EDTA-free tablets]). The cell suspension was transferred to a glass Dounce homogenizer and the cells were homogenized with 250 strokes at room temperature. The homogenate was spun at $1000 \times g$ for 10 min at 16 °C. The supernatants were centrifuged at $10,000 \times g$ for 10 min at 16 °C. The supernatants were spun again at $100,000 \times g$ for 35 min at 16 °C. The pellet,

representing the purified membrane fraction, was resuspended in 1× PBS to a final concentration of 14 mg of wet membrane per ml of 1× PBS. The membranes were then lysed in Amphipol A18 solubilization buffer (20 mM Tris-HCl [pH 8.0], 250 mM NaCl, 100 mM (NH$_4$)$_2$SO$_4$, 0.5% A18 and 1× protease inhibitor cocktail [Roche Complete EDTA-free tablets]) for 10 min at room temperature. The suspension was centrifuged for 25 min at 100,000 $\times$ g at 16 °C. The supernatant was incubated with a small volume of pre-equilibrated Ni-nitrilotriacetic acid (NTA) beads for 1.5 h on a rotating platform at room temperature. The mixture was transferred to an Eco-column (BioRad) and washed with 30 bed volumes of washing buffer (20 mM Tris-HCl [pH 8.0], 100 mM (NH$_4$)$_2$SO$_4$, 1 M NaCl, 30 mM imidazole). The Tri FPPR Env was eluted with 10 bed volumes of elution buffer (20 mM Tris-HCl [pH 8.0], 100 mM (NH$_4$)$_2$SO$_4$, 250 mM NaCl, 250 mM imidazole).

The uncleaved Tri FPPR Envs and gp41 stumps in the preparation were counterselected by incubation with 40 μg/ml of the 19b antibody and 40 μg/ml of the F240 antibody and 200 μl of Protein A-Sepharose beads at room temperature for 30 min. The mixture was applied to an Eco-Column (BioRad) and the eluate was incubated with 200 μl Protein A-Sepharose beads alone to remove residual antibodies.

### Cryo-EM specimen preparation
To prepare the purified Tri FPPR Env for single-particle cryo-EM analysis, the purified Env-A18-lipid complexes were dialyzed against a buffer containing 20 mM Tris-HCl (pH 8.0), 100 mM (NH$_4$)$_2$SO$_4$, 250 mM NaCl, and 10 μM BMS-806. Before cryo-plunging, Cymal-6 (Anatrace) was added to the Env solutions at a final concentration of 0.005%. A 3-μl drop of 0.3 mg/ml Env solution was applied to a glow-discharged UltrAufoil R1.2/1.3 300 mesh Gold grid (Electron Microscopy Sciences), blotted for 2 sec, and then plunged into liquid ethane and flash-frozen using an FEI Vitrobot Mark IV.

### Cryo-EM data collection
Before data collection, cryo-grids were first visually screened on a 200-kV Tecnai Arctica microscope (Thermo Fisher). Qualified grids were imaged in a 300-kV Titan Krios microscope (Thermo Fisher) equipped with a Gatan Bio-Quantum energy filter, at a nominal magnification of 105,000 times. Coma-free alignment and parallel illumination were manually optimized prior to data collection. Cryo-EM data for the Tri-FPPR Env were collected on the K3 Summit direct electron detector (Gatan) at a pixel size of 0.825 Å in a super-resolution counting mode, with an accumulated dose of 51.7 electrons/Å$^2$ across 47 frames per movie. With defocus ranging from –0.7 to –2.2 μm, a total of 19,534 movies were acquired semi-automatedly through SerialEM[62]. The collection process involved four steps: Global focusing and square selection, hole filtration, local focusing for each group of holes, and data acquisition.

### Cryo-EM data processing
The raw movies were first drift-corrected and dose-weighted at a super-resolution pixel size (0.825 Å) in the MotionCor2 program[63]. During the generation of micrographs, the first two frames in each raw movie were rejected for potentially high initial drift. Micrographs with poor quality due to being ice-contaminated or broken were manually screened and removed from the dataset using EMAN2 software[64]. The drift-corrected micrographs were then used for the determination of the precise defocus with the Gctf program[65]. The additional dose-weighted micrographs were sent for the following analysis. DeepEM, a deep learning-based particle extraction algorithm that helps recognize and select particles[66], was adopted to automate particle picking from the dose-weighted micrographs. After manual examination, 1,540,890 particles were available for further analysis.

2D classifications were conducted in ROME, a software that combines maximum likelihood-based image alignment and statistical manifold learning-based classification[67]. Picked particles were classified at 3.3 Å/pixel using a box size of 90 × 90 pixels. During the first two rounds of reference-free classification, junk particles were rejected based on the quality of the

averaged images of each 2D class, leaving 650,604 particles with diverse orientations for the following 3D analysis.

Particles derived from 2D classification were used to reconstruct an ab initio 3D model in RELION 3.0[68]. After lowpass filtering to 60 Å, this served as the initial model for 3D analysis. The following 3D classification and refinement were also performed in RELION 3.0, generally adopting the 'global to local' paradigm[68]. Setting the pixel size to 3.3 Å and 1.65 Å respectively, the first two rounds of 3D classification were both performed with global searching and conducted in two stages, i.e., in the initial 20 iterations, the HealPix order was set to 2 and the threshold for the probability calculation was set to 15 Å; the HealPix order was enhanced to 3 and the resolution limit was adjusted to 10 Å in the subsequent 25 iterations. Four classes were identified after global searching, among which 2 classes were favored for their apparent trimer shape and abundant structural details. Other particles from the leftover classes were checked and reclassified in 2D. The selected particles, containing 405,049 particles, were subjected to further auto-refinement. Local search was introduced in the midstream of auto-sampling when the HealPix order rose from 2 to 4.

Similar to the analysis conducted on the asymmetric AE2 Env, a pseudo-C3 symmetric operation was adopted to eliminate the anisotropic angular distribution along the central axis of the Tri FPPR Env trimer. In detail, the pseudo-C3 symmetric axis of the selected class was first aligned with the z-axis through a global auto-refinement, where a modified model with z-axis C3 symmetry served as the reference. Particles were rotated 120° around the z-axis twice in the same direction and then went through local auto-refinement with the HealPix order set to 4. The subsequent 3D classification on the expanded datasets was expected to generate 4 classes; among these 4 classes, 3 major classes corresponded to one conformation with each of the C3 symmetrical orientations, and the remaining class consisted of poorly resolved particles. One of the 3 major classes with better quality was sorted out and combined with the remaining single class for further analysis. After removing all the duplicated particles, the consistent and unexpanded dataset of 405,049 particles was subjected to another round of 3D classification with local searching, setting HealPix to 5 and angular sigma σ to 4 (the setting of σ indicates a local angular searching scope on all three Euler angles in the unit of the HealPix order). Six classes were thus generated; the major class consisting of 63% of the input dataset, i.e., 255,419 particles, was eventually selected based on satisfactory structural details in the Env ectodomain and recognizable density in the MPER region.

To extract out more details and evaluate possible heterogeneity, focused 3D classification was performed on the selected class. Focused 3D classification typically masks out the density of regions of lower interest to enhance the ability to discriminate finer structural details in areas of greater interest[69]. To achieve this goal, Chimera[70] was used to segment the map and subtract the density of the region of interest. RELION 3.0[68] was then utilized to generate the corresponding mask from the subtracted map and further add a cosine-shaped soft edge for smoothing. Masks were further aligned with experimental 2D images to generate modified 2D projections with CTF corrected and background noise added. The 3D model for the region of interest was rebuilt from the 2D projections and then used for a round of 3D classification that skipped orientational assignments. Specifically, such modified datasets were generated for the gp41 portion, along with the MPER region, and the gp120 portion. Reconstruction of one portion with the in-plane shift and Euler angles of each particle from focused classification was found to result in a blurry structure of the masked region; this may indicate potential relative motions between the gp120 subunits and gp41 subunits along with the MPER region. Different data processing procedures were therefore adopted to improve the outcome, as detailed below.

For the focused classification of the gp41 portion along with the MPER region, we employed two rounds of classification without alignment, accompanied by different preprocessing to ease the impact of the relative motions and balance the low signal-to-noise ratio at the same time. In detail, one classification was applied after the alignment on the whole trimer while another classification was instead applied after the alignment on the gp41 portion and MPER region only. With the regularization parameter T set to

10, each classification resulted in 10 classes after 50 iterations. The two rounds of classification both yielded 2 major classes out of 10 that demonstrated different pictures of the MPER region with limited quality. For the classification round that aligned on the whole trimer, we refer to these classes as the original class 3 and original class 6; for the classification round that aligned on the gp41 portion and the MPER region, we refer to these classes as class A and B. By counting the overlapped particles, the results from the two rounds of classification exhibited orthogonality, with a recognizable overlapping tendency between the classes. In detail, class A appeared to exhibit a preference for the original class 6 and not for the original class 3. Inspired by this cross-validation, we built the intersection of the original class 6 and class A and also the difference set of the original class 3 and class A (Supplementary Table 3). The latter set underwent another round of classification and yielded one major class. Eventually, two final classes, designated as Tri FPPR.1 and Tri FPPR.2, were attained from such cleaner datasets, which contained 57,088 particles and 50,646 particles, respectively. Tri FPPR.1 and Tri FPPR.2 not only maintained the asymmetric characteristics of the Env ectodomain from the original classes but also showed stronger and clearer density in the MPER region.

A round of auto-refinement was applied to the whole trimer of each class in the end. Two half-maps of each state with a pixel size of 0.825 Å were built. The masked resolutions reached 3.4 Å for Tri FPPR.1 and 3.5 Å for Tri FPPR.2, measured by gold-standard FSC at a 0.143-cutoff.

For the focused classification of the gp120 portion, C3 symmetry expansion was first applied as a data augmentation step, followed by the subtraction of a single protomer from the trimer. This approach facilitated a clearer classification of primary Env protomer conformations. Two rounds of focused classification were applied with the 'coarse to fine' pattern, where the first round focused on one protomer out of the trimer and the second one subtracted the gp120 V1/V2 region from the protomer for further analysis. Similar to the approach used for the gp41 portion (see above), alignment on the gp120 portion was applied before the second round of focused classification on the V1/V2 region. Two major classes for the gp120 V1/V2 region were classified out of ten after 50 iterations, resulting in two conformations of the gp120 subunit referred to as gp120.1 and gp120.2, which mainly differ in the V1/V2 region. After removing the duplicated expansion and reconstructing the whole trimer, 4 different combination patterns of gp120 V1/V2 classes were identified in the Env trimer, i.e., class (2,1,2) containing 26,633 particles; class (1,2,2) containing 14,023 particles; class (2,2,1) containing 25,921 particles; and class (2,2,2) containing 105,509 particles. The resulting four maps all reached a masked resolution higher than 5 Å, measured by gold-standard FSC at a 0.143-cutoff.

## Atomic model building and refinement

The asymmetric structure of the cleaved AE2.1 Env trimer with three BMS-806 molecules bound (PDB ID: 8FAE)[41] was used as a reference model to build the Tri FPPR.1 and Tri FPPR.2 Env structures. The fitting was further improved in Coot[71]. Env glycans were also manually refined in Coot with the 'Glycan' model, also with 8FAE as the reference. Additionally, the structure of the MPER chain was built on the Tri FPPR.2 map first because of its stronger density. The MPER fragment was then rigidly fitted in the lowpass-filtered Tri FPPR.1 Env map, along with minor refinements. Lowpass filtering the maps to a resolution of 5 Å helped to address ambiguities in the orientation of the MPER_N helix during model building due to the stronger density and the overall similarity of the MPER regions in the 5-Å maps of Tri FPPR.1 and Tri FPPR.2.

A hypothetical composite model was built based on the Tri FPPR.1 model and adjusted to accommodate the two lowpass-filtered Tri FPPR.1 and Tri FPPR.2 maps simultaneously. The composite model was constructed with minimal alterations to most parts of the Tri FPPR.1 model, with the majority of changes made to the MPER and its surrounding regions. Specifically, MPER segments resolved in the Tri FPPR.2 model were cropped and integrated into the Tri FPPR.1 model. By aligning the two 5 Å-filtered maps, helix-like fragments were rigidly fitted into the density and

incorporated, followed by refinement and moderate adjustment of the free loops between the helices to ensure structural coherence. Though the MPER of chain F is absent in both Tri FPPR.1 and Tri FPPR.2 models, the lowpass-filtered Tri FPPR.2 map contains an isolated density that is consistent with the chain F MPER$_N$ helix. Based on this density, chain F in the composite model was extended to include an MPER$_N$ helix, following the approach described above.

## Statistics and reproducibility

For the experiments shown in Fig. 5, a Mann-Whitney test was used to compare the LD/N values of cleaved and uncleaved Env structures. The relationships between the LD/N values and the resolved gp120 and gp41 termini were evaluated by Spearman rank order correlations with two-tailed P values. Purification of the Tri FPPR Env was performed successfully more than two times.

## Reporting summary

Further information on research design is available in the Nature Portfolio Reporting Summary linked to this article.

## Data availability

Data supporting the findings reported in this manuscript are available from the corresponding authors upon reasonable request. The cryo-EM density maps have been deposited in the Electron Microscopy Data Bank (EMDB) under accession codes EMD-61553 (Tri FPPR.1) and EMD-61554 (Tri FPPR.2). The model coordinates have been deposited in the RCSB Protein Data Bank (PDB) under accession codes 9JKF (Tri FPPR.1) and 9JKG (Tri FPPR.2). The PDB entries (2PV6, 7SC5, 8FAE, 7N6U, 8FAD, 4ZMJ, 6DLN, 7SKA, 6E8W, 4G6F, 1TJH and 8G8A) used for comparison are available in the PDB. EMDB entries (EMD-25022, EMD-21413 and EMD-25178) used in this study are available in the EMDB. Source data for the Fourier Shell Correlation (FSC) curves in Supplementary Fig. 3d and Supplementary Fig. 11b are provided in the file Supplementary Data 1.

## Code availability

All software and computational tools used in this study are publicly available. Data collection was carried out with SerialEM (version 3.6.11)[62]. Sample movement and drift were corrected using MotionCor2 (version 1.6.3)[63]. CTF determination was carried out with Gctf (version 1.06)[65]. Automated particle picking was conducted using DeepEM (version 1.0)[66]. Micrograph screening and verification of auto-picked particles were both performed using EMAN2 (version 2.91)[64], followed by 2D classification with ROME (version 1.1.2)[67] and 3D classification with RELION (version 3.0)[68]. During focused classification, masks were generated using UCSF Chimera (version 1.16)[70] and applied in RELION (version 3.0)[68] to constrain specific regions of interest. Model building, further refinement and glycan adjustments were performed in Coot (version 0.9.8)[71] and Phenix (version 1.18.2)[72]. Structural alignment and comparison were conducted in PyMOL (version 2.5.4, Schrödinger LLC), UCSF Chimera (version 1.16)[70] and ChimeraX (version 1.7.1)[73]. Local resolution was estimated using ResMap (version 1.19.2)[74]. Difference maps were calculated by UCSF Chimera (version 1.16)[70], with detailed parameters and procedures described in Supplementary Fig. 10. Structure figures were created with PyMOL (version 2.5.4, Schrödinger LLC), UCSF Chimera (version 1.16)[70] or ChimeraX (version 1.7.1)[73].

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

## Acknowledgements

We thank Ms. Elizabeth Carpelan for manuscript preparation. This study was supported by the National Natural Science Foundation of China (12125401), National Key Research and Development Program of China (2023YFF1204400 and 2023YFF1204401), the National Institutes of Health of the United States (AI124982, AI145547, AI060354 and AI178833), by the Michael Siff Fund for Basic Research and by a gift from the late William F. McCarty-Cooper. The research was also supported by the Center of Life Science at Peking University and the Basic Research Core of the University of Alabama, Birmingham Center for AIDS Research (NIH grant AI027767). Cryo-EM data were collected at the Harvard Cryo-EM Center for Structural Biology at Harvard Medical School. The data processing was performed in part in the Weiming No. 1 and Life Science No. 1 High-Performance Computing Platform at Peking University.

## Author contributions

S.Z., J.S., and Y.M. conceived this study. Z.Z., S.A., H.T.N., and J.S. designed and characterized the Tri FPPR Env. H.D. and J.C.K. generated the Env-expressing cells. S.Z. established a purification scheme for the Envs and prepared Env trimers. S.Z. analyzed cleavage, trimerization, and Env antigenicity. S.Z. screened the samples for optimal cryo-EM imaging. S.Z., Y.Q., and K.W. collected and pre-processed the cryo-EM data. Y.Q. and K.W. analyzed the cryo-EM data, refined the maps, and built the structural models. Y.Q., K.W., S.Z., J.S., and Y.M. analyzed the Env structural models. Y.Q., K.W., S.Z., Y.M., and J.S. wrote the manuscript with input from the other authors.

## Competing interests

The authors declare no competing interests.
