## [Transparent Peer Review file · Communications Biology]

The membrane-proximal external region of human immunodeficiency virus (HIV-1) envelope glycoprotein trimers in A18-lipid nanodiscs

Corresponding Author: Professor Youdong Mao

Version 0:

Reviewer comments:

Reviewer #1

(Remarks to the Author)

In the manuscript "The membrane-proximal external region of HIV-1 envelope glycoprotein trimers in amphipol-lipid nanodiscs" by Qi, Y., et al, cryo-EM structures of cleaved BMS-806-complexed HIV AD8 FPPR-stabilized trimers in amphipol-lipid nanodiscs are reported. Two main maps were obtained yielding two similar, although not identical, structural models termed Tri FPPR.1 and Tri FPPR.2. The authors predominantly focus their structural analysis on the membrane proximal external regions (MPER), the gp41 alpha-9 helices that lead into the MPERs, and the gp120 C-termini that are freed upon proteolytic cleavage of Env. Although average resolutions of the full structures are ~3.5 Å, within MPER the local resolutions range from 6-8 Å. Several features are observed in the structures, including asymmetry of protomers within each trimer, distinct positioning of gp41 alpha-9 helices and gp120 C-termini in the vicinity of the gp41 membrane proximal external regions (MPER), which notably vary across the different protomers. A composite model is generated between the two obtained structures, and the authors suggest that MPERs combined with alpha-9 helices and gp120 C-termini form a membrane proximal base that stabilizes the Env ectodomain.

The obtained structures represent an important contribution to the field, and complement existing structures of Env in lipid environments, including from the present authors. In many cases, however, hypothetical claims regarding the structures are presented as factual, empirically-based conclusions. For instance, in the abstract, statements such as: "A trimeric base composed of interlocking gp41 membrane-proximal external region (MPER) helices, gp120 C-termini and gp41 α 9 helices supports the metastable pretriggered Env", or "MPER conformational dynamics govern the tilt angles of α 9 helices, which in turn regulate loop-to-helix transitions in the gp41 heptad repeat (HR1N) regions", or "Helical HR1N transitions promote Env protomer opening and formation of the long HR1N-HR1C helix that translocates the fusion peptide to the target membrane", are all hypothetical interpretations of the structures that are presented here as facts. Such extensive intermingling of structural descriptions with presumed conclusive functional mechanisms, both in the abstract and elsewhere in the text, detracts from the structures themselves. This is further complicated by the fact that the Env at hand is a stabilized mutant (FPPR-based) and one in which the BMS-806 inhibitor is bound, making functional conclusions regarding wild type Env difficult.

Comments:

1. The model for conformational cascades is based on data presented in table 2 which compares the two FPPR structures against modified trimer AE2 structure. Comparisons against wt AD8 structure in this table would be helpful to assess possible effects of the stabilizing mutations.
2. Alternative models to the one in which asymmetry emanates from order to disorder transitions in MPER can apply to explain the causation of conformational cascades, but are not addressed.
3. The 6-8 Å local resolution range of MPER poses uncertainty, especially due to further disorder observed downstream of MPER.
4. A composite model of the MPER base region is stated as stabilizing the pre-triggered trimer, but no empirical evidence is

provided to address such stabilization.

5. Line 183, Figure 1: The statement “Thus, when Env is viewed down the trimer axis from the perspective of the membrane (as in the left panel of Fig. 1a), the FPPR-HR1N conformation on each gp41 protomer is related to the α 9-MPER structures on the counterclockwise protomer...” is unclear from the figure. Addition of labels and/or use of cartoon representation would help clarify this point.

6. Line 237: In the statement “By taking advantage of multiple interprotomer contacts, these relatively small Env elements combine to increase tensile strength and, along with the viral membrane, help to maintain Env trimer symmetry...”, such claims, including tensile strength, lack empirical evidence.

7. Line 244: Liberties taken to “tentatively model” aspects of the composite structure are concerning. Either the maps support a structural model or they do not.

8. Experimental amphipol-lipid nanodisc density is not addressed.

9. Line 401: Statement that the tilt of the alpha-9 helices is inversely related to the “strength of the connection” with MPER is subjective. Unclear how such strength is defined or measured in this context.

10. Generally, shifts in angles of the protomers are not easy to discern and in many cases not even shown in the figures or specified in the text.

Reviewer #2

(Remarks to the Author)

The authors structurally characterize a variant full-length Env from HIV-1AD8 containing several mutations and in the presence of ligand BMS-806, all of which have been shown to stabilize the closed pretriggered conformation (State 1) of the fusion glycoprotein in cell membranes. The structure resembles that of a related Env construct AE2 reported recently by the same group, with specific respect to the asymmetric arrangement of gp120 and gp41 subunits. New in the two trimer maps generated is the appearance of low-resolution density corresponding to the MPER region in one of the three gp41 subunits. The authors develop Env structural models that juxtaposes the MPER region with the gp120 C-terminus from the same protomer and the gp41 α 9 helix from an adjacent protomer, and use these to explain how MPER and gp120 C-terminal mutations might globally impact Env conformation. The authors combine the two solved structures to present a speculative model of the Env ectodomain base and use the results to postulate early structural changes that activate Env for membrane fusion. At the end, the authors present a minimal amount of data to demonstrate heterogeneity in the glycosylation pattern of the V1/V2 region at the Env apex.

The manuscript is a direct follow-up to the Reference 42 (Wang, et al, Comm Bio 2023), and there is minimal new information. The map of the MPER region is low resolution and it is unclear if it would be oriented as modeled or even if the ectodomain base of amphipol-solubilized Env mirrors that of Env in membranes (see below). Also, both the trimeric model of the ectodomain base and the model of early conformational changes is speculative at best and could do with additional biochemical or biophysical support.

Major points:

1. Biochemical data suggest the MPER interacts with the lipid bilayer, and it is reasonable to assume that its structure in the context of whole Env requires this interaction. It is unclear whether amphipol A18-solubilized Env has the requisite bilayer to characterize MPER structure accurately. Amphipols are polymers thought to directly belt transmembrane domains, solubilizing and stabilizing them. Amphipols are generally not good at solubilizing membranes, so any included lipids in the solubilized proteins are thought to be closely associated with the transmembrane domain. In reference 44 (which is now published online), the authors state in unshown data that the lipid composition of A18-solubilized Env matched that of Env in SMA-lipid nanoparticles, but it is unclear if that was in terms of relative lipid distribution or absolute lipid amount. It is telling that no bilayer could be visualized in the reconstructions. Could this be due to particle variability or bilayer flexibility, or could it be due to the very small amount of lipids in these particles? Gel filtration might be utilized to compare the size of A18- and SMA-particles to get a sense of absolute lipid content. But in the absence of data supporting the existence of a decent lipid bilayer in these particles, it remains hard to know whether the modeled MPER structure reflects what is really going on at the membrane.

2. The map corresponding to the MPER regions has a resolution of 8 angstroms (line 150), near the limit for accurate assignment of an alpha helix. Please show the model drawn into the original map and not the 5-angstrom filtered map in Figure 3. This low resolution precludes some overly-strong interpretations which are asserted as fact throughout the latter part of the manuscript, such as salt-bridges and cation- π interactions (lines 221-225, line 323, and later references).

3. There is not a lot of evidence to support the composite trimeric MPER model discussed in Figure 4. The authors acknowledge that there is a lot of clashing that is relieved by vertically displacing the MPER of the F subunit. In addition, the C-termini of the modeled MPERs appear to be too far apart to support association of the N-terminal regions of the TM domains. On the other hand, the vertical displacement might account for tilting of the Env 3-fold axis with respect to the membrane. But tilted species expose at least one MPER region (here, it would be the one on subunit F), and therefore would

be vulnerable to anti-MPER antibodies, much unlike state 1 stabilized Env.

4. The interpretation of the structural differences in protomers in terms of on-pathway allosteric shifts promoting Env activation/membrane fusion (Figure 6) could merely be a consequence of Env tilting with respect to the bilayer. Env bending away from protomer 3 might extend subunit F MPER (loss of helical structure), releasing the F a9 helix and E C-terminus which are now available to stabilize MPER segments in B and D and the HR1(N) helix in F. Asymmetries in interprotomer angle would be a consequence of this reorganization.

Minor points:

1. What data suggests that pretriggered Env in state 1 is 3-fold symmetric?

2. Line 131 – I would change “without imposing symmetry constraints” to “without imposing strict symmetry constraints.” The authors did apply C3 pseudosymmetry profiling and C3 expansion at points in their analyses (steps 3 and 6).

3. Lines 143-144 – “The C-terminal heptad repeat” generally refers to the gp41 HR2, which is the broken a9 helix in the native Env structure. HR1(C) should be referred as the C-terminal segment of the N-terminal heptad repeat.

4. Lines 299-300 – “HIV-1 neutralization by MPER-directed bNAbs occurs only after receptor binding.” This is not exactly correct. Evidence suggests Env fluctuates into an MPER accessible state with low frequency prior to CD4 binding, but CD4 binding stabilizes that state. See references DOI: 10.1084/jem.20101907 and DOI: 10.1371/journal.ppat.1010531 for irreversible inhibition of anti-MPER antibodies prior to HIV-1 introduction to cells.

5. Sodroski has shown that the Env population in cells can be very heterogeneous but that heterogeneity is much reduced in virus. How might this fact impact the gp120 heterogeneity observed in this study given that the Env was purified from a cell line.

Reviewer #3

(Remarks to the Author)

Qi and coworkers use cryo-EM of HIV-1 Env glycoproteins in amphipol-lipid nanodiscs to provide new structural information about the membrane-proximal external region (MPER) in the context of a largely intact, cleaved Env trimer. The MPER is a key determinant controlling Env-mediated membrane fusion and contains highly conserved neutralization epitopes that are accessible in neutralization-resistant primary isolates. The study shows that an asymmetric assembly of 3 MPERs form a base for the gp120-gp41 head domain and provide a model for how the MPERs regulate transitioning of gp41 to the prehairpin conformation upon CD4 activation of gp120-gp41. The work is convincing and the findings of the study will be of major interest to researchers studying HIV-1 Env structure, function and immunogenicity.

Optional suggestions:

An annotated linear map of Tri FPPR, color-coded according to panels a, b and c, added to the top of figure 1 would be useful to orient the reader. (Note: I found the FPPR designation a bit confusing given that the FPPR acronym also refers to the fusion peptide proximal region).

Several earlier studies have examined the role of the MPER in the HIV-1 Env membrane fusion mechanism. There is also a plethora of information regarding the interaction between bNAbs and the MPER. The authors could discuss the findings of these studies with respect to the structural context of the MPER elucidated here.

Version 1:

Reviewer comments:

Reviewer #1

(Remarks to the Author)

The authors have sufficiently addressed previous concerns, and I have no further comments.

Reviewer #3

(Remarks to the Author)

The authors have addressed my suggestions in the revised manuscript and it should be accepted for publication.

Reviewer #4

(Remarks to the Author)

This very interesting manuscript provides important structural data and interpretation that addresses longstanding questions about HIV Env structural transitions and the specific role of the mysterious MPER region (with major implications for antigen

design). The data is presented clearly with appropriate caveats. Although in some cases the results are more hypothesis-generating than definitive conclusions, the manuscript presents the data and interprets it in the context of prior structures in an appropriate manner for such a complex problem (making significant progress over prior reports). These data will be of broad interest to the Env field and are likely to inspire future functional studies. The authors were highly responsive to previous reviewer critiques, and the manuscript is significantly improved and more clear as a result. In my opinion, this manuscript is now suitable for publication.

Point-by-point response to the referees' comments

We thank the reviewers for their careful reading of our manuscript and for their suggestions to improve the manuscript. We have responded to the reviewers' suggestions as follows:

Reviewer #1

In the manuscript "The membrane-proximal external region of HIV-1 envelope glycoprotein trimers in amphipol-lipid nanodiscs" by Qi, Y., et al, cryo-EM structures of cleaved BMS-806-complexed HIV AD8 FPPR-stabilized trimers in amphipol-lipid nanodiscs are reported. Two main maps were obtained yielding two similar, although not identical, structural models termed Tri FPPR.1 and Tri FPPR.2. The authors predominantly focus their structural analysis on the membrane proximal external regions (MPER), the gp41 alpha-9 helices that lead into the MPERs, and the gp120 C-termini that are freed upon proteolytic cleavage of Env. Although average resolutions of the full structures are ~3.5 Å, within MPER the local resolutions range from 6-8 Å. Several features are observed in the structures, including asymmetry of protomers within each trimer, distinct positioning of gp41 alpha-9 helices and gp120 C-termini in the vicinity of the gp41 membrane proximal external regions (MPER), which notably vary across the different protomers. A composite model is generated between the two obtained structures, and the authors suggest that MPERs combined with alpha-9 helices and gp120 C-termini form a membrane proximal base that stabilizes the Env ectodomain.

The obtained structures represent an important contribution to the field, and complement existing structures of Env in lipid environments, including from the present authors. In many cases, however, hypothetical claims regarding the structures are presented as factual, empirically-based conclusions. For instance, in the abstract, statements such as: "A trimeric base composed of interlocking gp41 membrane-proximal external region (MPER) helices, gp120 C-termini and gp41 α 9 helices supports the metastable pretriggered Env", or "MPER conformational dynamics govern the tilt angles of α 9 helices, which in turn regulate loop-to-helix transitions in the gp41 heptad repeat (HR1N) regions", or "Helical HR1N transitions promote Env protomer opening and formation of the long HR1N-HR1C helix that translocates the fusion peptide to the target membrane", are all hypothetical interpretations of the structures that are presented here as facts. Such extensive intermingling of structural descriptions with presumed conclusive functional mechanisms, both in the abstract and elsewhere in the text, detracts from the structures themselves.

Response: We are pleased that the reviewer recognized the importance of these Env structures to the field. In our effort to place the structures into a larger context concisely, we melded the Results and Discussion. This can create difficulties for the reader to separate the observations from the hypothetical framework. Therefore, we have revised the Abstract (with the updated text located in lines 4-15) and separated the Results and Discussion, carefully indicating throughout the manuscript sections that are hypothetical.

This is further complicated by the fact that the Env at hand is a stabilized mutant (FPPR-based) and one in which the BMS-806 inhibitor is bound, making functional conclusions regarding wild type Env difficult.

Response: Although the Tri FPPR mutant is significantly more resistant to soluble CD4, CD4-mimetic compounds and cold inactivation than the wild-type HIV-1_{AD8} Env, indicative of stabilization of the pretriggered conformation, it mediates cell-cell fusion and virus entry at 71% and 17%, respectively, of the wild-type HIV-1_{AD8} Env levels (Z. Zhang *et al.*, *iScience* 27: 110141

(2024)). When solubilized in A18, the Tri FPPR Env is recognized by broadly neutralizing antibodies (bNAbs) but not by poorly neutralizing antibodies (pNAbs), as is the case for cleaved cell-surface Env (S. Zhang *et al.*, *J Virol* 98: e0063124 (2024)). This native pattern of antigenicity is observed in both the absence and presence of BMS-806, which has been shown to stabilize a pretriggered (State-1) Env conformation (J Munro *et al.*, *Science* 346: 759-763 (2014); H.T. Nguyen *et al.*, *J Virol* 97: e0185722 (2023)). To demonstrate the relatedness of the Tri FPPR Env structure to the wild-type HIV-1_{AD8} Env structure, we have added data on the wild-type HIV-1_{AD8} Env to Table 2 and Supplementary Figures 5, 6 and 9.

Comments:

1. The model for conformational cascades is based on data presented in table 2 which compares the two FPPR structures against modified trimer AE2 structure. Comparisons against wt AD8 structure in this table would be helpful to assess possible effects of the stabilizing mutations.

Response: This is an excellent suggestion. We have added the data on the wild-type HIV-1_{AD8} Env to Table 2 and Supplementary Figures 5, 6 and 9. Although the wt AD8 trimer is slightly less compact than the stabilized Env trimers, its overall features are similar to those of the other Envs, including the Tri FPPR Envs.

2. Alternative models to the one in which asymmetry emanates from order to disorder transitions in MPER can apply to explain the causation of conformational cascades, but are not addressed.

Response: Asymmetry in the Env trimer could exist even in the absence of receptor binding or membrane disruption, and we mention this possibility in the revised manuscript (lines 284-288 and lines 733-734).

3. The 6-8 Å local resolution range of MPER poses uncertainty, especially due to further disorder observed downstream of MPER.

Response: The degree of disorder in the MPER and TM regions is higher than that in the rest of the Tri FPPR Env ectodomain. Nonetheless, each of the Tri FPPR maps exhibits traceable density in the MPER. The MPER density in the Tri FPPR.1 and Tri FPPR.2 maps is similar, even though the most stable density in each map arises from difference protomers. In the revised manuscript, we show the Tri FPPR.2 MPER map at higher resolution (Figure 4b). We limit our conclusions to those that are justified by the resolution achieved.

4. A composite model of the MPER base region is stated as stabilizing the pre-triggered trimer, but no empirical evidence is provided to address such stabilization.

Response: Single amino acid substitutions in multiple MPER residues result in loss of integrity of the HIV-1 Env pretriggered conformation. In the revised manuscript, Figure 4b shows the location of these residues on the modeled MPER. The results are consistent with the importance of the MPER_N helix, which abuts the gp120 C-terminus and gp41 α9 helix of the adjacent protomer, to the stability of the pretriggered (State-1) conformation (PTC).

5. Line 183, Figure 1: The statement “Thus, when Env is viewed down the trimer axis from the perspective of the membrane (as in the left panel of Fig. 1a), the FPPR-HR1N conformation on each gp41 protomer is related to the α9-MPER structures on the counterclockwise protomer...”

is unclear from the figure. Addition of labels and/or use of cartoon representation would help clarify this point.

Response: We have added labeled images to Figure 1a and added Supplementary Figure 6b to clarify this point.

6. Line 237: In the statement “By taking advantage of multiple interprotomer contacts, these relatively small Env elements combine to increase tensile strength and, along with the viral membrane, help to maintain Env trimer symmetry...”, such claims, including tensile strength, lack empirical evidence.

Response: We have removed the statement from the revised manuscript.

7. Line 244: Liberties taken to “tentatively model” aspects of the composite structure are concerning. Either the maps support a structural model or they do not.

Response: Although the MPER density from chain F of the Tri FPPR.2 map is poorly ordered compared with the chain B MPER, the lowpass-filtered map density associated with the chain F MPER_N helix does allow placement of this helix in the composite structure. We avoid any conclusions requiring higher resolution. The modified text can be found in lines 615-618 and 679-681. Also, we removed the rest of the chain F MPER trace from the composite structure and deleted any analysis based on these C-terminal MPER residues; accordingly, Supplementary Table 2 and the corresponding text (lines 324-336) have been modified.

8. Experimental amphipol-lipid nanodisc density is not addressed.

Response: In the revised manuscript, we show the amphipol-lipid nanodisc density for the Tri FPPR.1 and Tri FPPR.2 maps in Supplementary Figure 8. The tilting of the Env ectodomain with respect to the nanodisc axis positions some of the Env protomers closer to the nanodisc. Of interest, a higher degree of order in the MPER density is associated with the protomer closer to the nanodisc. This relationship is discussed in the revised manuscript (lines 187–196).

9. Line 401: Statement that the tilt of the alpha-9 helices is inversely related to the “strength of the connection” with MPER is subjective. Unclear how such strength is defined or measured in this context.

Response: We have reworded this statement to clarify that the tilt of the alpha-9 helices is inversely related to the degree of order in the associated MPER density (lines 264-273 of the revised manuscript).

10. Generally, shifts in angles of the protomers are not easy to discern and in many cases not even shown in the figures or specified in the text.

Response: We have added indicators of the opening angles in the figures or figure legends, where appropriate.

Reviewer #2

The authors structurally characterize a variant full-length Env from HIV-1AD8 containing several mutations and in the presence of ligand BMS-806, all of which have been shown to stabilize the

closed pretriggered conformation (State 1) of the fusion glycoprotein in cell membranes. The structure resembles that of a related Env construct AE2 reported recently by the same group, with specific respect to the asymmetric arrangement of gp120 and gp41 subunits. New in the two trimer maps generated is the appearance of low-resolution density corresponding to the MPER region in one of the three gp41 subunits. The authors develop Env structural models that juxtaposes the MPER region with the gp120 C-terminus from the same protomer and the gp41 a9 helix from an adjacent protomer, and use these to explain how MPER and gp120 C-terminal mutations might globally impact Env conformation. The authors combine the two solved structures to present a speculative model of the Env ectodomain base and use the results to postulate early structural changes that activate Env for membrane fusion. At the end, the authors present a minimal amount of data to demonstrate heterogeneity in the glycosylation pattern of the V1/V2 region at the Env apex.

The manuscript is a direct follow-up to the Reference 42 (Wang, et al, Comm Bio 2023), and there is minimal new information. The map of the MPER region is low resolution and it is unclear if it would be oriented as modeled or even if the ectodomain base of amphipol-solubilized Env mirrors that of Env in membranes (see below). Also, both the trimeric model of the ectodomain base and the model of early conformational changes is speculative at best and could do with additional biochemical or biophysical support.

Response: Despite its functional importance, there currently are no detailed structures of the unliganded MPER in the context of the complete Env trimer. In the sole published report (S. Yang *et al.*, *Nat Commun* 13:6393 (2022)), only a few unstructured MPER amino acid residues were modeled in the cryo-EM structure of an Env trimer, and the modeled MPER segment was based on the NMR structure of an MPER peptide in DPC micelles. In this manuscript, we provide two maps of HIV-1 Env trimers that allow complete traces of the MPER in one protomer and partial traces in other protomers. Importantly, the densities of the MPERs and associated structures in the two maps are consistent, justifying our proposal of a composite model for the membrane-proximal base of Env. In the revised manuscript, the contribution of the MPER to the maintenance of the pretriggered Env conformation is supported by structure-function analysis of the MPER mutants, which are mapped on the derived structural model (Figure 4b). Although more work on the native MPER structure needs to be done, our structures provide an initial perspective on the MPER in the context of an Env trimer.

The reviewer correctly points out that the model of early conformational changes is based on certain assumptions. As in our response to Reviewer 1, in the revised manuscript, we separate the Results and Discussion, clearly distinguishing the structures from hypotheses.

Major points:

1. Biochemical data suggest the MPER interacts with the lipid bilayer, and it is reasonable to assume that its structure in the context of whole Env requires this interaction. It is unclear whether amphipol A18-solubilized Env has the requisite bilayer to characterize MPER structure accurately. Amphipols are polymers thought to directly belt transmembrane domains, solubilizing and stabilizing them. Amphipols are generally not good at solubilizing membranes, so any included lipids in the solubilized proteins are thought to be closely associated with the transmembrane domain. In reference 44 (which is now published online), the authors state in unshown data that the lipid composition of A18-solubilized Env matched that of Env in SMA-lipid nanoparticles, but it is unclear if that was in terms of relative lipid distribution or absolute lipid amount. It is telling that no bilayer could be visualized in the reconstructions. Could this be due to particle variability or bilayer flexibility, or could it be due to the very small amount of lipids in

these particles? Gel filtration might be utilized to compare the size of A18- and SMA-particles to get a sense of absolute lipid content. But in the absence of data supporting the existence of a decent lipid bilayer in these particles, it remains hard to know whether the modeled MPER structure reflects what is really going on at the membrane.

Response: Amphipol A18 can directly solubilize integral membrane proteins, including the HIV-1 Env, from membranes. The distribution of lipids in A18-lipid nanodiscs and SMA-lipid nanodiscs was similar. In the revised manuscript, we include low-contour images that show the relationship of the Env ectodomain to the amphipol-lipid nanodisc (Supplementary Figure 8). The lipids in the nanodiscs are not sufficiently ordered to allow conclusions about the membrane lipids and their potential impact on MPER structure. However, as discussed in our response to Reviewer 1, there is a relationship between the proximity of the MPER protomers to the nanodisc and the degree of order in the MPER density. The implications of this relationship are discussed below and in the revised manuscript (lines 187–196). Finally, we note that previous attempts to reconstitute detergent-solubilized Env trimers into proteoliposomes failed to resolve either the MPERs or membranes (K. Rantalainen *et al.*, *Cell Rep* 31: 107583 (2020)). As the particles in these cryo-EM studies are aligned by the better-ordered Env ectodomains, the structure of the less-well-ordered membrane-proximal components are generally not defined.

2. The map corresponding to the MPER regions has a resolution of 8 angstroms (line 150), near the limit for accurate assignment of an alpha helix. Please show the model drawn into the original map and not the 5-angstrom filtered map in Figure 3. This low resolution precludes some overly-strong interpretations which are asserted as fact throughout the latter part of the manuscript, such as salt-bridges and cation-pi interactions (lines 221-225, line 323, and later references).

Response: As discussed in our response to Reviewer 1, in Figure 4b of the revised manuscript, we show the fit of the MPER model to the higher-resolution map density. In the revised manuscript, we have removed all conclusions dependent on resolutions higher than those achieved. The updated text can be found in lines 181-182. Predictions related to the formation of specific bonds have been removed from Figure 3 of the revised manuscript.

3. There is not a lot of evidence to support the composite trimeric MPER model discussed in Figure 4. The authors acknowledge that there is a lot of clashing that is relieved by vertically displacing the MPER of the F subunit. In addition, the C-termini of the modeled MPERs appear to be too far apart to support association of the N-terminal regions of the TM domains. On the other hand, the vertical displacement might account for tilting of the Env 3-fold axis with respect to the membrane. But tilted species expose at least one MPER region (here, it would be the one on subunit F), and therefore would be vulnerable to anti-MPER antibodies, much unlike state 1 stabilized Env.

Response: The rationale for assembling a composite MPER structure is the high degree of similarity between the MPER structures resolved in the Tri FPPR.1 and Tri FPPR.2 maps. With respect to the asymmetric Env ectodomain, the best resolved MPER structures are found in different protomers and are therefore complementary. Details of the methods used to assemble the composite structure are found in lines 607-618 and in the Figure 4 legend (lines 674-681) of the revised manuscript. We have eliminated the segment of the F subunit that is the least well ordered from the composite model. We agree that Env tilting with respect to the nanodisc is one explanation for the observed MPER structures. We show that the MPER structures brought closest to the nanodisc by Env tilting exhibit the most ordered density (Supplementary Figure 8). We agree that Env tilting to a degree that exposes one of the MPERs to antibody binding may

coincide with disruption of the pretriggered (State-1) Env conformation.

4. The interpretation of the structural differences in protomers in terms of on-pathway allosteric shifts promoting Env activation/membrane fusion (Figure 6) could merely be a consequence of Env tilting with respect to the bilayer. Env bending away from protomer 3 might extend subunit F MPER (loss of helical structure), releasing the F a9 helix and E C-terminus which are now available to stabilize MPER segments in B and D and the HR1(N) helix in F. Asymmetries in interprotomer angle would be a consequence of this reorganization.

Response: The commonality among multiple HIV-1 Env trimers solubilized from membranes suggests that the observed asymmetric trimers represent a preferred conformation. In the revised manuscript, we discuss alternative models consistent with the available evidence. In one model, this preferred Env conformation is an intermediate on the virus entry pathway (lines 389-396 of the revised manuscript). Other potential explanations of Env asymmetry exist and are discussed, including the tilting of Env suggested by the reviewer (See lines 284-288). We note that these models are not mutually exclusive.

Minor points:

1. What data suggests that pretriggered Env in state 1 is 3-fold symmetric?

Response: Recent studies on the stoichiometry of Env changes that stabilize or destabilize the pretriggered Env conformation suggest that conformational symmetry among the Env protomers contributes to the maintenance of the pretriggered Env. This submitted manuscript, which is available in bioRxiv, is cited in our revised manuscript. However, this evidence is indirect and does not rule out structural differences among the Env protomers. In the revised manuscript, we discuss different models, leaving open the possibility that the pretriggered (State-1) Env is asymmetric (lines 397-404).

2. Line 131 – I would change “without imposing symmetry constraints” to “without imposing strict symmetry constraints.” The authors did apply C3 pseudosymmetry profiling and C3 expansion at points in their analyses (steps 3 and 6).

Response: The goal of the pseudo-C3 symmetry expansion in step 3 of our cryo-EM workflow (Supplementary Figure 2) is to ensure that the particles have an equal chance of being assigned to each of the main projection angles relative to the trimer axis, allowing us to select the most accurate result and detect subtle asymmetries. In step 6, C3 expansion is employed as a data augmentation technique to aid in classifying the primary classes of Env protomer conformation, rather than a means of imposing symmetry constraints on the trimer. The unique combinations of protomer conformations within the trimer were reconstructed by only one representative set of particles after de-duplication.

3. Lines 143-144 – “The C-terminal heptad repeat” generally refers to the gp41 HR2, which is the broken a9 helix in the native Env structure. HR1(C) should be referred as the C-terminal segment of the N-terminal heptad repeat.

Response: In lines 103-104 of the revised manuscript, we have reworded this sentence to eliminate any ambiguity.

4. Lines 299-300 – “HIV-1 neutralization by MPER-directed bNAbs occurs only after receptor binding.” This is not exactly correct. Evidence suggests Env fluctuates into an MPER accessible

state with low frequency prior to CD4 binding, but CD4 binding stabilizes that state. See references DOI: 10.1084/jem.20101907 and DOI: 10.1371/journal.ppat.1010531 for irreversible inhibition of anti-MPER antibodies prior to HIV-1 introduction to cells.

Response: We have modified this statement to indicate that, for primary HIV-1 strains, neutralization by MPER bNAbs occurs after receptor binding. The updated text can be found in Lines 36 and 340 of the revised manuscript. As the reviewer notes, fluctuations of Env into an MPER antibody-accessible state occur only at a low frequency. Ruprecht *et al.*, *JEM* 2011 show that MPER bNAbs can neutralize primary viruses before target cell engagement, but this requires more than 10 hours of incubation of the virus-antibody mixture! Under these extended incubation conditions, several pNAbs also neutralize virus. Thus, Envs of primary HIV-1 do spontaneously sample conformations downstream of the pretriggered conformation, but these transitions occur very slowly and inefficiently. Schapiro *et al.*, *PLoS Pathogens* 2022 show that the laboratory-adapted HIV-1_{HXBc2} can be neutralized by MPER bNAbs efficiently. However, a chimeric Env that exhibits phenotypes indicative of a stabilized pretriggered conformation is more resistant to MPER bNAbs. These results indicate that the pretriggered (State-1) conformation of primary HIV-1 Envs is only rarely accessible to MPER-directed bNAbs.

5. Sodroski has shown that the Env population in cells can be very heterogeneous but that heterogeneity is much reduced in virus. How might this fact impact the gp120 heterogeneity observed in this study given that the Env was purified from a cell line.

Response: Much of the conformational heterogeneity of the cell-associated Env results from the abundance of uncleaved Env, which is flexible. Most of the uncleaved Tri FPPR Env in the cell lysate was removed by counterselection with a mixture of pNAbs. Thus, ~96% of the Env in the preparation used for cryo-EM is cleaved and antigenically similar to the cleaved Env on the virion surface. We note that the amount of uncleaved Env in our imaged preparation is insufficient to account for the observed frequencies of the gp120.1 and gp120.2 classes.

Reviewer #3

Qi and coworkers use cryo-EM of HIV-1 Env glycoproteins in amphipol-lipid nanodiscs to provide new structural information about the membrane-proximal external region (MPER) in the context of a largely intact, cleaved Env trimer. The MPER is a key determinant controlling Env-mediated membrane fusion and contains highly conserved neutralization epitopes that are accessible in neutralization-resistant primary isolates. The study shows that an asymmetric assembly of 3 MPERs form a base for the gp120-gp41 head domain and provide a model for how the MPERs regulate transitioning of gp41 to the prehairpin conformation upon CD4 activation of gp120-gp41. The work is convincing and the findings of the study will be of major interest to researchers studying HIV-1 Env structure, function and immunogenicity.

Optional suggestions:

An annotated linear map of Tri FPPR, color-coded according to panels a, b and c, added to the top of figure 1 would be useful to orient the reader. (Note: I found the FPPR designation a bit confusing given that the FPPR acronym also refers to the fusion peptide proximal region).

Response: We have added an annotated linear map of the Tri FPPR Env to Supplementary Figure 1b. FPPR in our Env nomenclature refers to three changes in the fusion peptide-proximal region (A532V/I535M/L543Q) that stabilize the pretriggered (State-1) Env conformation. This is

explained in the legend to the figure.

Several earlier studies have examined the role of the MPER in the HIV-1 Env membrane fusion mechanism. There is also a plethora of information regarding the interaction between bNAbs and the MPER. The authors could discuss the findings of these studies with respect to the structural context of the MPER elucidated here.

Response: Within the available space constraints, we discuss the implications of our results to Env-mediated membrane fusion and antibody-MPER interaction. The updated text is located in lines 337-341 and 389-410 in the revised manuscript.

We thank the reviewers for their helpful suggestions and trust that the revised manuscript is now suitable for publication in *Communications Biology*.